# RPWithPrior: Label Differential Privacy in Regression

## Abstract

With the wide application of machine learning techniques in practice, privacy preservation has gained increasing attention. Protecting user privacy with minimal accuracy loss is a fundamental task in the data analysis and mining community. In this paper, we focus on regression tasks under $\epsilon$-label differential privacy guarantees. Some existing methods for regression with $\epsilon$-label differential privacy, such as the RR-On-Bins mechanism and its variant, discretized the output space into finite bins and then applied randomized response (RR) algorithms. To efficiently determine these finite bins, the authors rounded the original responses down to integer values. However, such operations does not align well with real-world scenarios. To overcome these limitations, we model both original and randomized responses as *continuous* random variables, avoiding discretization entirely. Our novel approach estimates an optimal interval for randomized responses and introduces new algorithms designed for scenarios where a prior is either known or unknown. Additionally, we prove that our algorithm, RPWithPrior, guarantees $\epsilon$-label differential privacy. Numerical results demonstrate that our approach gets better performance compared with the Gaussian, Laplace, Staircase, and RRonBins, Unbiased mechanisms on the Communities and Crime, Criteo Sponsored Search Conversion Log, California Housing datasets and some simulated datasets.

## 1 Introduction

Due to the widespread application of machine learning methods for model training, there has been a significant increase in attention towards the privacy of individual data in recent years. To address this concern and protect the privacy of data used for training, the concept of differential privacy was introduced as a popular notion by Dwork et al. (2006b).

Differential privacy (DP) aims to quantify the level of privacy protection provided by measuring the differences between two neighboring datasets and the privacy guarantees of a given mechanism. In the context of $\epsilon$-DP, the parameter $\epsilon$ is commonly referred to as a privacy parameter or privacy budget. It serves as a knob to adjust the degree of privacy. Smaller values of $\epsilon$ correspond to stronger privacy guarantees, indicating a higher level of protection for individual data.

A particularly important class of differential privacy is known as label differential privacy (label DP). In label DP, the sensitive information lies in the labels, while the features are considered public and non-sensitive. For formal definitions of DP and Label DP, we refer the reader to the preliminary section in Appendix A.

In this work, we focus on Label DP. A canonical example of label DP is online advertising, where a user's profile and the contextual information of an ad are considered public features, while the ad conversion (the label or response) remains private. Additionally, demographic surveys can also be framed as label DP problems. In a census survey, the demographic information such as gender or age is considered public, while the income is deemed sensitive.

In the context of DP or label DP for classification problems, the labels are typically drawn from a finite set. However, in regression tasks, the response variable may not necessarily be limited to a finite number of values and could even be uncountable. Some existing methods for achieving $\epsilon$-label differential privacy in regression, such as the RR-On-Bins mechanism (Ghazi et al., 2022) and its variant (Badanidiyuru et al., 2023), relied on discretizing the output space into finite bins and applying

RR algorithms, similar to approaches used in classification tasks with $\epsilon$-label DP. To determine finite bins, the RR-On-Bins mechanism and its variant used a dynamic programming approach, which could be resource-intensive in terms of storage requirements. Furthermore, computing optimal bins involved in these methods could be highly time-consuming. To save storage space and computational time, the authors discretized the labels by rounding the original responses down to integer values. The methods of Ghazi et al. (2022) and Badanidiyuru et al. (2023) fundamentally convert a regression problem into a classification problem within the framework of Label Differential Privacy.

In this paper, we introduce a novel approach to tackle these challenges of label differential privacy in regression from a fundamentally distinct perspective. Unlike existing methods (Ghazi et al., 2022; Badanidiyuru et al., 2023) that discretized the output space into finite bins for randomized response, our approach models both original and randomized responses directly as continuous random variables, eliminating discretization-induced errors. We derive an optimal interval for randomized responses and propose a new algorithm designed to maximize the probability of preserving neighborhood relationships between original and randomized responses and guarantee $\epsilon$-label differential privacy.

Our contributions can be summarized as follows:

1. **To the best of our knowledge, this work introduces continuous random variables for non-additive mechanisms in label DP.** We propose RPWithPrior, a novel label DP mechanism for regression tasks. The core idea of RPWithPrior is to estimate an optimal interval for randomized responses. We compute optimal values for $A_1$ and $A_2$ using prior information to determine the optimal interval $[A_1 - \zeta, A_2 + \zeta]$ in which randomized responses are constrained (Lemma 3.5 and Algorithm 1). Here, $\zeta > 0$ is a tunable parameter controlling the interval's expansion. If the input (original) response $y$ falls within this interval $[A_1, A_2]$, it is uniformly mapped to its neighboring region $\mathcal{N}_y = [y - \zeta, y + \zeta]$ with a certain probability, while uniformly assigning the remaining probability to the remaining interval $[A_1 - \zeta, A_2 + \zeta] \setminus \mathcal{N}_y$ (see Equation (3.4)). If the original responses lie outside $[A_1, A_2]$, satisfying the label differential privacy is sufficient to generate valid randomized responses (see Remark 3.4). Algorithm 2, named as 'RPWithPrior', provides the details to obtain randomized responses. Furthermore, we provide a theoretical guarantee that RPWithPrior satisfies $\epsilon$-label differential privacy.

2. **We expand upon the RPWithPrior algorithm to address the scenario where the global prior distribution is unknown, which is also guarantees for $\epsilon$-label DP.** Initially, we estimate the global prior distribution using the histogram of the data randomized by Laplace mechanism. Then, we train a regression model on the randomized responses which are drawn from the RPWithPrior algorithm.

3. **Our proposed method is both efficient and effective**. Our method has lower time complexity, making it more user-friendly for big data computations.

To assess the performance of our method, we conducted experiments on three datasets: the Communities and Crime dataset (Redmond and Baveja, 2002), the Criteo Sponsored Search Conversion Log dataset (Tallis and Yadav, 2018), and the California housing dataset (Pace and Barry, 1997).

We validate the effectiveness of our proposed method by measuring the test mean squared error (MSE) and compare with the Gaussian mechanism, Laplace mechanism, and Staircase mechanism and RRonBins mechanism. Our experimental results consistently reveal that our method achieves the lower error rate in average and standard deviation, highlighting its superior performance.

To validate the efficiency, we also provide a comparison of the computational complexity, including the runtime and unique number of samples or discretized samples in the training process for the aforementioned mechanisms. In addition, we provide a comprehensive comparison between some existing methods for regression and ours in Appendix J.

The remainder of this paper is structured as follows: Section 2 reviews related works on differential privacy and label differential privacy in both classification and regression tasks. Section 3 presents our proposed method for achieving label differential privacy in regression with prior known, followed by the method for label differential privacy in regression with prior unknown in Section 4. Numerical implementations are discussed in Section 5. Finally, we conclude with our findings and highlight the limitations of this work in Section 6.

## 2 RELATED WORKS

Numerous methods exist for designing DP algorithms, including output or objective perturbation (Chaudhuri et al., 2011), DP versions of SGD (Song et al., 2013), functional mechanism (Zhang et al., 2012), DP-SGD with norm clipping and noise adding (Abadi et al., 2016), among others. One classical method predating the notion of differential privacy is RR (Warner, 1965), which was applied to inputs from a finite set. In the RR algorithm, sensitive inputs are mapped to themselves with a certain probability and uniformly to other values with the remaining probability. Unfortunately, the RR algorithm suffers from a large error (Chan et al., 2012). Consequently, a significant number of works have been proposed to improve accuracy, often by relaxing to weaker privacy models (Acharya et al., 2020; Cheu et al., 2019; Erlingsson et al., 2019). The second class of differential privacy methods is the additive noise mechanisms. Popular mechanisms include the Gaussian mechanism (Dwork et al., 2006a; Smith et al., 2018; 2021), Laplace mechanism (Dwork et al., 2006b), staircase mechanism (Geng and Viswanath, 2014), and others, which add noise to adjacent inputs to calibrate sensitivity. Over the last decade, DP algorithms have been integrated with machine (deep) learning, applying to adversarial learning (Phan et al., 2020), multiple parties (Shokri and Shmatikov, 2015), language models (McMahan et al., 2017) and knowledge transfer (Papernot et al., 2016). For convenience, open-source libraries that integrate machine learning frameworks have been developed, such as TensorFlow Privacy and PyTorch Opacus.

For label Differential privacy, either classification or regression, numerous researchers have focused on this topic. In order to address the label differential privacy problem in classification, Ghazi et al. (2021) proposed the Label Privacy Multi-Stage Training (LP-MST) algorithm by calibrating the raw user data using the 'RRWithPrior' method, where the prior is predicted by the previous model.

Furthermore, Malek Esmaeili et al. (2021) presented two approaches, namely PATE and ALIBI, based on the PATE framework and the Laplace mechanism, specifically designed for label-privacy machine learning settings in classification. In the case of PATE, they utilized the PATE framework for semi-supervised learning, achieving an excellent tradeoff between empirical privacy loss and accuracy. On the other hand, ALIBI applied additive Laplace noise to a one-hot encoding of the label and then performed Bayesian inference using the prior to denoise the mechanism's output. They demonstrated that ALIBI improves upon the work of Ghazi et al. (2021) in high-privacy regimes. In Esfandiari et al. (2022), the authors presented new mechanisms for label differential privacy via clustering. Their approach involved clustering training examples based on non-private feature vectors, randomly resampling labels from examples within the same cluster, and producing a training set with noisy labels alongside a modified loss function. They demonstrated that if the clusters were large and of high quality, the model minimizing the modified loss converged to a small excess risk at a rate similar to non-private learning.

For regression tasks, one popular approach involves the additive noise mechanisms (Balle and Wang, 2018; Fernandes et al., 2021; Geng and Viswanath, 2014). A recent method is the RR-on-Bins mechanism (Ghazi et al., 2022), which discretized the output space into finite bins and applied the RR algorithm to these bins. The finite bins were determined using a dynamic programming approach. Another notable method is the optimal unbiased randomizers for regression with label differential privacy (Badanidiyuru et al., 2023), which built upon the RR-on-Bins mechanism by introducing an unbiased constraint to enhance its performance.

## 3 LABEL DP FOR REGRESSION WITH PRIOR KNOWN

In this section, our focus is on regression with label differential privacy. We consider a dataset for regression denoted as $(x, y) \in (\mathcal{X}, \mathcal{Y})$, drawn from an unknown distribution. The regression task is to learn a predictor $f_\theta : \mathcal{X} \to \mathcal{Y}$ by minimizing a loss function $\mathcal{L}(f_\theta(x), y)$. The loss function, denoted as $\mathcal{L}(\cdot, \cdot)$, is typically used for regression tasks, such as mean squared error. Our goal is to design a randomized method that maps original responses to randomized responses by maximizing the probability of preserving neighborhood relationships, while ensuring $\epsilon$-label differential privacy.

In Ghazi et al. (2022) and Badanidiyuru et al. (2023), the authors assumed that the randomized outputs were sampled from a finite set. In contrast, we consider both the original and randomized response variables as continuous random variables. We denote the original response variable as $Y$ and the corresponding randomized output variable as $\tilde{Y}$.

Let us consider the concept of $\epsilon$-label differential privacy. We have the following proposition:

**Proposition 3.1.** *Let $\mathcal{A} : \mathcal{Y} \to \tilde{\mathcal{Y}}$ be a randomized algorithm and $f_{\tilde{Y}|Y}(\tilde{y}, y)$ represent the conditional probability density function conditioned on $Y = y$. $\mathcal{A}$ satisfying $\epsilon$-label DP is equivalent to the following inequality:*

$$\int_{y_0}^{\tilde{y}} f_{\tilde{Y}|Y}(\tilde{y}, y) d\tilde{y} \leq e^{\epsilon} \int_{y_0}^{\tilde{y}} f_{\tilde{Y}|Y}(\tilde{y}, y') d\tilde{y}, \ \forall y, y' \in \mathcal{Y}, \forall (y_0, \tilde{y}) \subseteq \tilde{\mathcal{Y}}. \tag{3.1}$$

*When taking the derivative of both sides of* (3.1) *with respect to $\tilde{y}$, we obtain $f_{\tilde{Y}|Y}(\tilde{y}, y) \leq e^{\epsilon} f_{\tilde{Y}|Y}(\tilde{y}, y')$ holds for all $y, y' \in \mathcal{Y}$ and $\tilde{y} \in \tilde{\mathcal{Y}}$ almost everywhere.*

**Remark 3.2.** In the following, we characterize $\epsilon$-label DP as follows:

$$f_{\tilde{Y}|Y}(\tilde{y}, y) \leq e^{\epsilon} f_{\tilde{Y}|Y}(\tilde{y}, y'), \ \forall y, y' \in \mathcal{Y}, \forall \tilde{y} \in \tilde{\mathcal{Y}}. \tag{3.2}$$

Next, we formulate an optimization model to address regression problems under $\epsilon$-label differential privacy with a known prior, as discussed in Subsection 3.1, using a fixed set $\mathcal{I}$. An optimal set $\mathcal{I}$ is determined in Subsection 3.2. Once the optimal set is established, a randomized algorithm is introduced, and the $\epsilon$-label DP property of the proposed algorithm is ensured in Subsection 3.3.

## 3.1 MODEL FORMULATION

In this subsection, we assume we know the distribution of original responses of regression and the corresponding probability density function is $f_Y(y)$. Let $f_{\tilde{Y}|Y}(\tilde{y}, y)$ represent the conditional probability density function conditioned on $Y = y$ and $\mathcal{N}_y = [y - \zeta, y + \zeta]$ be a $\zeta$-neighborhood of $y$, where $\zeta$ is a positive number.

Our method is motivated by the empirical distribution of the responses. Since most responses are concentrated within a finite interval, with only a few extreme outliers, we seek an optimal interval $\mathcal{I}$. This interval maximizes the probability that a response $y$ is mapped to a randomized value $\tilde{y}$ within its neighborhood $\mathcal{N}_y$, for all $y \in \mathcal{I}$, while still satisfying $\epsilon$-label differential privacy.

In this paper, we consider a subset $\mathcal{I} = [A_1, A_2] \subseteq (-\infty, +\infty)$ (determine in Subsection 3.2) and assume that $f_{\tilde{Y}|Y}(\tilde{y}, y)$ is a positive constant for any $y \in \mathcal{I}$ and $\tilde{y} \in \mathcal{N}_y$. Define $\mathcal{N}_{\mathcal{I}} = [A_1 - \zeta, A_2 + \zeta]$, our goal is to find a randomized algorithm mapping the original responses, whether in $\mathcal{I}$ or not, to the set $\mathcal{N}_{\mathcal{I}}$. This algorithm must ensure the $\epsilon$-label DP property while maximizing the probability of transitioning from $y$ to $\tilde{y} \in \mathcal{N}_y$ for all $y \in \mathcal{I}$. Then the optimization model is

$$\max \int_{\mathcal{I}} f_Y(y) \left[ \int_{\mathcal{N}_y} f_{\tilde{Y}|Y}(\tilde{y}, y) d\tilde{y} \right] dy$$

$$\text{s.t.} \begin{cases} \int_{-\infty}^{+\infty} f_{\tilde{Y}|Y}(\tilde{y}, y) d\tilde{y} = 1, \forall y \in \mathcal{I}, \\ f_{\tilde{Y}|Y}(\tilde{y}, y) \leq e^{\epsilon} f_{\tilde{Y}|Y}(\tilde{y}, y'), \forall y, y' \in \mathcal{I}, \forall \tilde{y} \in \mathcal{N}_{\mathcal{I}}, \\ f_{\tilde{Y}|Y}(\tilde{y}, y) = v, \forall y \in \mathcal{I}, \forall \tilde{y} \in \mathcal{N}_y, \\ f_{\tilde{Y}|Y}(\tilde{y}, y) \geq 0, \forall \tilde{y}, y, \end{cases} \tag{3.3}$$

where the first and fourth constraints in (3.3) are from the properties of conditional probability density function, the second constraint in (3.3) is from (3.2) and the third constraint is from the assumption $f_{\tilde{Y}|Y}(\tilde{y}, y), \forall y \in \mathcal{I}, \forall \tilde{y} \in \mathcal{N}_y$ is a positive constant, denoted as $v$.

The following lemma shows the maximizer and the maximal value of the objective function in (3.3).

**Lemma 3.3.** *Let $\mathcal{I} = [A_1, A_2]$, $\zeta$ be a positive constant and $\epsilon > 0$ be a privacy budget. Define $\gamma = 2\zeta + e^{-\epsilon}(A_2 - A_1)$, $\mathcal{N}_y = [y - \zeta, y + \zeta], \forall y \in \mathcal{I}$ and $\mathcal{N}_{\mathcal{I}} = \cup_{y \in \mathcal{I}} \mathcal{N}_y$. For any $y \in \mathcal{I}$, the maximizer of* (3.3) *is*

$$f_{\tilde{Y}|Y}(\tilde{y}, y) = \begin{cases} 1/\gamma, & \text{if } \tilde{y} \in \mathcal{N}_y, \\ e^{-\epsilon}/\gamma, & \text{if } \tilde{y} \in \mathcal{N}_{\mathcal{I}}/\mathcal{N}_y, \\ 0, & \text{otherwise}, \end{cases} \tag{3.4}$$

*and the maximal value of the objective function is*

$$\int_{\mathcal{I}} f_Y(y) \left[ \int_{\mathcal{N}_y} f_{\tilde{Y}|Y}(\tilde{y}, y) d\tilde{y} \right] dy \leq 2\frac{\zeta}{\gamma} \int_{\mathcal{I}} f_Y(y) dy = 2\frac{\zeta}{\gamma} \int_{A_1}^{A_2} f_Y(y) dy \triangleq F(A_1, A_2).$$

*Proof.* We defer the proof to Appendix B. $\qquad\qquad\qquad\qquad\qquad\qquad\qquad\qquad\square$

It is important to note that the optimization model (3.3) can only provide $f_{\tilde{Y}|Y}(\tilde{y}, y)$ for all $y \in \mathcal{I}$. For $y \in (-\infty, +\infty) \setminus \mathcal{I}$, an $\epsilon$-label DP property is also required. We summarize $f_{\tilde{Y}|Y}(\tilde{y}, y)$ for all $y$ in Remark 3.4.

**Remark 3.4.** In this paper, our goal is to find a randomized algorithm mapping original responses, whether in $\mathcal{I}$ or not, to the set $\mathcal{N}_{\mathcal{I}}$. To guarantee the $\epsilon$-label DP property and combine (3.4), the conditional probability density function $f_{\tilde{Y}|Y}(\tilde{y}, y), \forall y \notin \mathcal{I}$ requires to satisfy

$$\max_{\tilde{y} \in \mathcal{N}_{\mathcal{I}}} (f_{\tilde{Y}|Y}(\tilde{y}, y)) \leq \frac{1}{\gamma}, \ \min_{\tilde{y} \in \mathcal{N}_{\mathcal{I}}} (f_{\tilde{Y}|Y}(\tilde{y}, y)) \geq \frac{e^{-\epsilon}}{\gamma}, \ f_{\tilde{Y}|Y}(\tilde{y}, y) = 0, \forall \tilde{y} \notin \mathcal{N}_{\mathcal{I}}, \int_{\tilde{y} \in \mathcal{I}} f_{\tilde{Y}|Y}(\tilde{y}|y) = 1.$$
$$(3.5)$$

One possible choice of $f_{\tilde{Y}|Y}(\tilde{y}, y), \forall y$ is

$$f_{\tilde{Y}|Y}(\tilde{y}, y) = \begin{cases} 1/\gamma, & \text{if } \tilde{y} \in \mathcal{N}_{P_{\mathcal{I}}(y)}, \\ e^{-\epsilon}/\gamma, & \text{if } \tilde{y} \in \mathcal{N}_{\mathcal{I}}/\mathcal{N}_{P_{\mathcal{I}}(y)}, \\ 0, & \text{otherwise.} \end{cases} \quad P_{\mathcal{I}}(y) = \begin{cases} A_2, & y > A_2, \\ y, & y \in \mathcal{I}, \\ A_1, & y < A_1. \end{cases} \quad (3.6)$$

Note that $\epsilon \geq 0$, then $e^{-\epsilon} \leq 1$ and $\gamma = 2\zeta + e^{-\epsilon}(A_2 - A_1) \leq 2\zeta + (A_2 - A_1)$, $e^{\epsilon}\gamma = 2\zeta e^{\epsilon} + (A_2 - A_1) \geq 2\zeta + (A_2 - A_1)$. Therefore, $e^{-\epsilon}/\gamma = 1/(e^{\epsilon}\gamma) \leq 1/(2\zeta + (A_2 - A_1)) \leq 1/\gamma$. Hence

$$f_{\tilde{Y}|Y}(\tilde{y}, y) = \begin{cases} 1/\gamma, & \text{if } y \in \mathcal{I}, \tilde{y} \in \mathcal{N}_y, \\ e^{-\epsilon}/\gamma, & \text{if } y \in \mathcal{I}, \tilde{y} \in \mathcal{N}_{\mathcal{I}}/\mathcal{N}_y, \\ \frac{1}{2\zeta + (A_2 - A_1)}, & \text{if } y \notin \mathcal{I}, \tilde{y} \in \mathcal{N}_{\mathcal{I}}, \\ 0, & \text{otherwise.} \end{cases} \quad (3.7)$$

Equations (3.6) and (3.7) present two options for $f_{\tilde{Y}|Y}(\tilde{y}, y)$ as discussed in Remark 3.4. Other choices are acceptable as long as the condition (3.5) is satisfied.

### 3.2 THE OPTIMAL INTERVAL

For a given interval $[A_1, A_2]$, Lemma 3.3 gives the maximum achievable value $F(A_1, A_2)$ of the optimization problem (3.3). Our goal is to find the optimal interval by maximizing $F(A_1, A_2)$ over all possible $A_1$ and $A_2$. Next, we estimate an optimal choice for the interval $\mathcal{I} = [A_1, A_2]$ based on an available prior.

In practice, the commonly used technique for estimating a prior is a histogram, which corresponds to a piecewise constant function. Here we assume that $f_Y(y)$ is a step function with nodes $n_0 < n_1 < \cdots < n_k$ and analyze the scenarios in which $A_1$ and $A_2$ fall into these $k + 2$ intervals with $A_1 \leq A_2$, respectively.

The following lemma aims to identify the maximizers $(A_1, A_2)$ of the objective function $F(A_1, A_2)$.

**Lemma 3.5.** *Suppose the support of $f_Y(y)$ consists of $k$ intervals with endpoints $n_0 < n_1 < \cdots < n_k$ and*

$$f_Y(y) = \begin{cases} \alpha_i, & y \in [n_i, n_{i+1}), i = 0, \cdots, k-2, \\ \alpha_{k-1}, & y \in [n_{k-1}, n_k], \\ 0, & \text{otherwise.} \end{cases}$$

*Let $\zeta$ be a positive constant, $\gamma = 2\zeta + e^{-\epsilon}(A_2 - A_1)$ and $F(A_1, A_2) = \frac{2\zeta}{\gamma} \int_{A_1}^{A_2} f_Y(y) dy$, where $F(A_1, A_2)$ is the maximum achievable value of the optimization problem (3.3) (derived in Lemma 3.3). When $A_1 \in (-\infty, n_0)$, $F(A_1, A_2)$ is increasing about $A_1$. When $A_2 \in (n_k, \infty)$, $F(A_1, A_2)$ is decreasing about $A_2$. When $A_1 \in [n_i, n_{i+1}]$ and $A_2 \in [n_j, n_{j+1}]$ with $i \leq j$,*

$$\nabla F_{A_1}(A_1, A_2) = \frac{-2\zeta}{\gamma^2} (e^{-\epsilon}(\alpha_i - \alpha_j)A_2 + c_1), \nabla F_{A_2}(A_1, A_2) = \frac{2\zeta}{\gamma^2} (e^{-\epsilon}(\alpha_i - \alpha_j)A_1 + c_2),$$

*and the possible critical points are $A_1 = e_1 = \frac{c_2}{e^{-\epsilon}(\alpha_j - \alpha_i)}$ and $A_2 = e_2 = \frac{c_1}{e^{-\epsilon}(\alpha_j - \alpha_i)}$, where $h = \alpha_i n_{i+1} + \sum_{\ell=i+1}^{j-1} \alpha_\ell (n_{\ell+1} - n_\ell) - \alpha_j n_j$, $c_1 = 2\zeta\alpha_i - e^{-\epsilon}h$, $c_2 = 2\zeta\alpha_j - e^{-\epsilon}h$.*

*Proof.* The proof defers to Appendix C. □

---

**Algorithm 1** Compute Optimal Interval

---

**Require:** A response $n_0 < n_1 < \cdots < n_k$, $\alpha_0, \cdots, \alpha_{k-1}$.
1:  Set $f = 0$.
2:  **for** $i = 0$ to $k - 1$ **do**
3:      **for** $j = i$ **to** $k - 1$ **do**
4:          $l_1 = n_i$, $l_2 = n_{i+1}$, $m_1 = n_j$ and $m_2 = n_{j+1}$,
5:          **if** $f < \max_{p,q=1,2}(F(l_p, m_q))$ **then**          ▷ Check $A_1 = n_i$ or $n_{i+1}$, $A_2 = n_j$ or $n_{j+1}$.
6:              $f = \max_{p,q=1,2}(F(l_p, m_q))$,
7:              $[A_1, A_2] = \arg\max_{p,q=1,2}(F(l_p, m_q))$.
8:          **end if**
9:          Compute possible critical points $A_1 = e_1$, $A_2 = e_2$.
10:         **if** $n_j < e_2 < n_{j+1}$ **then**                                              ▷ if $e_2 \in [n_j, n_{j+1}]$
11:             **if** $f < \max(F(n_i, e_2), F(n_{i+1}, e_2))$ **then**          ▷ check $A_1 = n_i$ or $n_{i+1}$, $A_2 = e_2$
12:                 $f = \max(F(n_i, e_2), F(n_{i+1}, e_2))$,
13:                 $[A_1, A_2] = \arg\max(F(n_i, e_2), F(n_{i+1}, e_2))$.
14:             **end if**
15:         **end if**
16:         **if** $n_i < e_1 < n_{i+1}$ **then**                                              ▷ if $e_1 \in [n_i, n_{i+1}]$
17:             **if** $f < \max(F(e_1, n_j), F(e_1, n_{j+1}))$ **then**          ▷ check $A_1 = e_1$, $A_2 = n_j$ or $n_{j+1}$
18:                 $f = \max(F(e_1, n_j), F(e_1, n_{j+1}))$,
19:                 $[A_1, A_2] = \arg\max(F(e_1, n_j), F(e_1, n_{j+1}))$.
20:             **end if**
21:         **end if**
22:         **if** $n_j < e_2 < n_{j+1}$ **and** $n_i < e_1 < n_{i+1}$ **then**   ▷ if $e_2 \in [n_j, n_{j+1}]$ and $e_1 \in [n_i, n_{i+1}]$,
23:             **if** $f < F(e_1, e_2)$ **then**                                              ▷ check $A_1 = e_1$, $A_2 = e_2$
24:                 $A_1 = e_1$, $A_2 = e_2$.
25:             **end if**
26:         **end if**
27:     **end for**
28: **end for**
29: **return** $A_1$ and $A_2$.

---

From Lemma 3.5, we find $F(A_1, A_2)$ increases when $A_1 \in (-\infty, n_0)$ and $F(A_1, A_2)$ decreases when $A_2 \in (n_k, +\infty)$, so the maximizer must be in the interval $[n_0, n_k]$. We turn to enumerate the $k$ intervals in $[n_0, n_k]$. When $A_1 \in [n_i, n_{i+1}]$ and $A_2 \in [n_j, n_{j+1}]$ with $i \leq j$, the possible critical points are $A_1 = e_1$ and $A_2 = e_2$.

To determine the maximizer, it is sufficient to evaluate the endpoints and points where the partial derivatives are zero. It is important to note that we will only check the value of $e_1$ if it falls within the interval $[n_i, n_{i+1}]$. Similarly, we will only check the value of $e_2$ if it falls within the interval $[n_j, n_{j+1}]$. The algorithm to compute optimal interval is described in Algorithm 1.

In Algorithm 1, the memory complexity is $O(k)$ because we only need to store a constant number of variables, including $n_0, \cdots, n_k$, $\alpha_0, \cdots, \alpha_{k-1}$, $A_1$, $A_2$, $f$, $e_1$ and $e_2$, where $k$ represents the number of intervals with piecewise constants in the support. The time complexity of Algorithm 1 is at most $O(k^2)$. In each iteration, we need to evaluate the function $F$, which requires $O(1)$ operations. We iterate over $i = 0, \cdots, k - 1$ and $j = i, \cdots, k - 1$. Generally, $k$ is not large, which can be controlled by users. Therefore, the algorithm for computing the optimal interval is efficient.

### 3.3 RESPONSE PRIVACY WITH PRIOR

Once the optimal interval $\mathcal{I}$ is chosen, the randomized output can be assigned to a specific point within $\mathcal{N}_{\mathcal{I}}$ according to the conditional probability density function (3.4), if the input response falls

within this interval $\mathcal{I}$. Otherwise, we sample according to the conditional probability density function $f_{\tilde{Y},Y}(\tilde{y}, y)$ satisfying (3.5). The whole procedure is presented in Algorithm 2, named as *RPWithPrior*, to sample the randomized response $\tilde{y}$.

---

**Algorithm 2** Response Privacy with Prior (RPWithPrior$_\epsilon$)

---

**Require:** A response $y$, a positive value $\zeta$, privacy budget $\epsilon$.
  1: Compute the optimal $A_1$, $A_2$ by Algorithm 1.
  2: **if** $y \in [A_1, A_2]$ **then**
  3:     Sample $\tilde{y}$ from the distribution with the conditional probability density function (3.4),
  4: **else**
  5:     Sample $\tilde{y}$ according to $f_{\tilde{Y}|Y}(\tilde{y}, y)$ satisfying (3.5).
  6: **end if**
  7: **return** A randomized response $\tilde{y}$.

---

Finally, we will show *RPWithPrior$_\epsilon$ is $\epsilon$-label DP*, where the proof defers to Appendix D.

# 4    LABEL DP FOR REGRESSION WITH PRIOR UNKNOWN

Our approach, RPWithPrior, is based on the availability of priors that are known in advance. However, there are scenarios where obtaining such priors is not always feasible. In this section, we propose an alternative algorithm that uses a histogram to approximate the prior $f_Y(y)$, computed from randomized responses generated by an additive mechanism (e.g., the Laplace mechanism) to guarantee label DP.

Let's consider the training dataset denoted as $D$ and $\mathcal{M}_{\epsilon,\text{Lap}}(y)$ be the randomized response of $y$ by Laplace mechanism for any $(x, y) \in D$. To be precise, we first calculate the sample expectation $\mu$: $\mu = \frac{1}{\#(D)} \sum_{(x,y) \in D} \mathcal{M}_{\epsilon,\text{Lap}}(y)$, where $\#(\cdot)$ is the element number in the set. For a given positive parameter $\sigma$, let $k_0$ and $k_1$ be integers satisfying $\mu + k_0\sigma \leq \min_{(x,y) \in D} \mathcal{M}_{\epsilon,\text{Lap}}(y) < \mu + (k_0 + 1)\sigma$ and $\mu + k_1\sigma < \max_{(x,y) \in D} \mathcal{M}_{\epsilon,\text{Lap}}(y) \leq \mu + (k_1 + 1)\sigma$, the nodes are set as $n_0 = \min_{(x,y) \in D} \mathcal{M}_{\epsilon,\text{Lap}}(y)$, $n_i = \mu + (i + k_0)\sigma, i = 1, \cdots, k_1 - k_0$ and $n_{k_1-k_0+1} = \max_{(x,y) \in D} \mathcal{M}_{\epsilon,\text{Lap}}(y)$. Define $[n] = \{0, \cdots, n\}$, $\forall n \in \mathbb{N}$ and $J_k = [n_k, n_{k+1})$, if $k \in [k_1 - k_0 - 1]$, and $J_k = [n_k, n_{k+1}]$, if $k = k_1 - k_0$, then the histogram expression of $f_Y(y)$ is

$$f_Y(y) = \left\{ \begin{array}{ll} \frac{\#(S_k)}{\#(D)}, & y \in J_k, \forall k \in [k_1 - k_0], \\ 0, & \text{otherwise}, \end{array} \right. \quad S_k = \{y | \mathcal{M}_{\epsilon,\text{Lap}}(y) \in J_k, (x, y) \in D\}.$$

Due to space constraints, we present the histogram algorithm (Algorithm 4) in Appendix G.

**Remark 4.1.** The length of the histogram intervals in Algorithm 4 can be adjusted according to specific requirements by modifying the value of $\sigma$. When the prior distribution is complex, a smaller interval length enables the histogram to more accurately approximate the prior distribution; however, this comes at the cost of increased computational complexity. In this paper, we set $\sigma$ to be the standard deviation of the randomized data by Laplace mechanism.

By utilizing the histogram as the prior, we can employ Algorithm 2 to obtain the optimal values of $A_1$ and $A_2$. These optimal values are then used to sample for each $(x, y)$ in the dataset $D$. Subsequently, we train the model $M$ on the datasets. To provide a comprehensive overview, we present the complete algorithm in Algorithm 3. Algorithm 3 is $(\epsilon_1 + \epsilon_2)$-label DP (see Appendix E). We also provide error analysis of the predicted responses and the the true responses, which is deferred to Theorem F in Appendix F, due to page limitations.

# 5    NUMERICAL IMPLEMENTATION

Our focus is on three specific datasets: the Communities and Crime dataset (Redmond and Baveja, 2002), the Criteo Sponsored Search Conversion Log dataset (Tallis and Yadav, 2018), and the California housing dataset (Pace and Barry, 1997). In the following experiments, we conduct a total

---

**Algorithm 3** Response Privacy with Histogram

---

**Require:** Dataset $D = \{(x_1, y_1), (x_2, y_2), \cdots, (x_n, y_n)\}$ and privacy budgets $\epsilon_1, \epsilon_2$.

1: Set $\tilde{D} = \emptyset$,
2: estimate the prior by histogram via $\text{Hist}_{\epsilon_1}$ in Algorithm 4,
3: **for** $(x_i, y_i) \in D$ **do**
4:      sample $\tilde{y}_i$ according to $\text{RPWithPrior}_{\epsilon_2}$ Algorithm 2,
5:      $\tilde{D} = \tilde{D} \cup \{(x_i, \tilde{y}_i)\}$,
6: **end for**
7: $M$ is the regression model trained on the set $\tilde{D}$.
8: **return** $M$.

---

Table 1: Comparison results on the Communities and Crime dataset.

| Privacy Budget | Laplace Mean ± Std | Gaussian Mean ± Std | Staircase Mean ± Std | RRonBins Mean ± Std | Ours Mean ± Std | Unbiased Mean ± Std |
|---|---|---|---|---|---|---|
| 0.05 | 8.3956 ± 3.1062 | 33.0362 ± 11.6948 | 7.0269 ± 3.0227 | 0.1033 ± 0.0119 | **0.0679 ± 0.0070** | 0.1293±0.0096 |
| 0.1 | 2.4534 ± 1.1036 | 17.5311 ± 7.1242 | 2.3541 ± 0.6848 | 0.0964 ± 0.0243 | **0.0683 ± 0.0061** | 0.1194±0.0056 |
| 0.3 | 0.3232 ± 0.0631 | 2.4218 ± 1.1848 | 0.2973 ± 0.0731 | 0.1107 ± 0.0221 | **0.0604 ± 0.0092** | 0.1110±0.0094 |
| 0.5 | 0.1615 ± 0.0513 | 0.8934 ± 0.3259 | 0.1127 ± 0.0226 | 0.1108 ± 0.0349 | **0.0544 ± 0.0065** | 0.1037±0.0089 |
| 0.8 | 0.0679 ± 0.0122 | 0.4376 ± 0.1317 | **0.0621 ±0.0124** | 0.1068 ± 0.0188 | **0.0453 ± 0.0057** | 0.0892±0.0064 |
| 1.0 | **0.0533 ± 0.0117** | 0.3138 ± 0.1096 | **0.0467 ± 0.0064** | 0.0962 ± 0.0133 | **0.0411 ± 0.0042** | 0.0826±0.0039 |
| 1.5 | **0.0353 ± 0.0081** | 0.1481 ± 0.0481 | **0.0344 ± 0.0055** | 0.0827 ± 0.0162 | **0.0372 ± 0.0023** | 0.0668±0.0041 |
| 2 | **0.0275 ± 0.0032** | 0.0947 ± 0.0167 | **0.0255 ± 0.0027** | 0.0628 ± 0.0110 | **0.0329± 0.0036** | 0.0521±0.0032 |
| 3 | **0.0223 ± 0.0018** | 0.0600 ± 0.0132 | **0.0241 ± 0.0026** | 0.0332 ± 0.0062 | **0.0244 ± 0.0027** | 0.0355±0.0021 |
| 4 | **0.0216 ± 0.0020** | 0.0368 ± 0.0084 | **0.0239 ± 0.0033** | 0.0282 ± 0.0038 | **0.0212 ± 0.0033** | **0.0259±0.0024** |
| 6 | **0.0188 ± 0.0017** | 0.0291 ± 0.0042 | 0.0249 ± 0.0025 | **0.0202± 0.0027** | 0.0205 ± 0.0023 | 0.0222±0.0016 |
| 8 | **0.0201 ± 0.0021** | 0.0231 ± 0.0020 | 0.0241 ± 0.0024 | **0.0187 ± 0.0020** | 0.0209 ± 0.0018 | 0.0204 ±0.0017 |
| $+\infty$ | **0.0182 ± 0.0016** | **0.0182 ± 0.0016** | **0.0182 ± 0.0016** | **0.0184 ± 0.0009** | **0.0177 ± 0.0021** | **0.0194 ±0.0018** |

of 10 trials for the Gaussian, Laplace, Staircase, RRonBins mechanisms and our proposed mechanism. In each trial, we randomly divide the entire dataset into training (80%) and test (20%) sets using different random seeds. The numerical implementations are available at `https://github.com/anonymousaabb/Regression_Privacy`.

For all regression tasks in our experiments, we employ mean squared error (MSE) as the loss function $\mathcal{L}$ and evaluate performance using both the mean and standard deviation of the test MSE. The reported results are the average and standard deviation of the test mean squared error over the 10 trails. Note that we evaluate results based on the interval ranging from mean minus standard deviation to mean plus standard deviation. The best-performing intervals are those that overlap with the interval of the smallest mean, which are highlighted in bold. Due to space constraints, descriptions of the datasets, technical details, and computational complexity analyses are provided in Appendix H. We also evaluate our proposed algorithm on simulated datasets (linear and nonlinear), which is deferred to Appendix I.

### 5.1 THE COMMUNITIES AND CRIME DATASET

In this subsection, we evaluate our proposed mechanism and compare with the Gaussian, Laplace, Staircase, RRonBins (Ghazi et al., 2022), and Unbiased (Badanidiyuru et al., 2023) mechanisms on the Communities and Crime dataset. Table 1 provides the quantitative comparison results of the Gaussian, Laplace, Staircase, RRonBins, Unbiased mechanisms and our proposed one for various privacy budgets.

Based on the results presented in Table 1, our proposed method exhibits smaller mean and standard deviation of the test MSE values across different privacy budgets compared to the other methods. Consequently, we can conclude that our proposed method has good performance compared with the additive noise mechanisms and the RRonBins, Unbiased mechanisms.

### 5.2 THE CRITEO SPONSORED SEARCH CONVERSION LOG DATASET

In the following, we quantitatively evaluate our proposed method. We compare our method with the Gaussian, Laplace, Staircase, RRonBins, and Unbiased mechanisms. Table 2 presents the quantitative comparison results for the Gaussian, Laplace, and Staircase mechanisms at different privacy budgets.

Table 2: Comparison results on the Criteo Sponsored Search Conversion Log dataset.

| Privacy Budget | Laplace Mean ± Std | Gaussian Mean ± Std | Staircase Mean ± Std | RRonBins Mean ± Std | Ours Mean ± Std | Unbiased Mean ± Std |
|---|---|---|---|---|---|---|
| 0.05 | 1878491.63 ± 376368.64 | 16803328.94 ± 998367.25 | 1736473.22 ± 122674.30 | 11339.71 ± 36.45 | **6882.55 ± 40.96** | 10572.59±1134.12 |
| 0.1 | 472390.91 ± 14030.15 | 4458946.92 ± 385131.83 | 464807.11 ± 19835.24 | 11328.04 ± 36.34 | **6839.72 ± 25.78** | 7884.54±664.82 |
| 0.3 | 60599.02 ± 1124.63 | 567545.99 ± 8905.76 | 55470.30 ± 1296.68 | 11185.20 ± 36.10 | 6526.58 ± 54.49 | **5807.11±393.60** |
| 0.5 | 24659.75 ± 745.69 | 215649.49 ± 2694.33 | 21717.01 ± 417.43 | 10907.33 ± 36.54 | 6302.61 ± 6246 | **4826.07±250.91** |
| 0.8 | 11982.80 ± 290.64 | 87173.02 ± 1582.45 | 10328.00 ± 255.31 | 10256.37 ± 37.39 | 5931.67±81.26 | **4769.94±210.42** |
| 1.0 | 8854.12 ± 267.56 | 58036.92 ± 1286.23 | 7762.13 ± 230.03 | 9744.08 ± 37.59 | 5742.74 ± 114.96 | **4630.83±171.56** |
| 1.5 | 5759.41 ± 178.68 | 28740.18 ± 481.94 | 5301.81 ± 274.90 | 8406.88 ± 36.57 | 4659.06 ± 72.15 | **4472.40±92.37** |
| 2 | 4787.05 ± 127.27 | 17620.97 ± 493.14 | **4516.25 ± 62.59** | 7294.93 ± 34.03 | **4306.80 ± 131.89** | 4459.39±78.33 |
| 3 | **3848.77 ±184.59** | 9780.89 ± 136.51 | 4103.87 ± 101.25 | 5577.50 ± 31.75 | **4042.74 ± 111.63** | 4330.81±58.60 |
| 4 | **3551.65 ± 164.53** | 7081.23 ± 259.26 | 4080.65 ± 128.19 | 4769.61 ± 25.01 | **3229.45 ± 129.65** | 4244.34±76.74 |
| 6 | **3339.38 ± 141.50** | 5024.29 ± 340.55 | 4144.02 ± 75.71 | 4371.68 ± 25.31 | **3185.96 ± 69.52** | 3241.17±44.89 |
| 8 | 3164.46 ±83.76 | 4163.70 ± 119.34 | 4322.13 ± 187.16 | 4333.12 ± 31.94 | 3104.52 ± 42.10 | **2911.97±58.3** |
| $+\infty$ | 3119.31 ± 99.39 | 3119.31 ± 99.39 | 3119.31 ± 99.39 | 4319.86 ± 29.27 | 3119.01 ± 70.31 | **2798.53±29.11** |

Based on the results presented in Table 2, our method and the unbiased mechanism outperform other mechanisms in the Criteo Sponsored Search Conversion Log dataset.

## 5.3 THE CALIFORNIA HOUSING DATASET

Table 3: Comparison results for Gaussian, Laplace and Staircase mechanisms on the Housing dataset.

| Privacy Budget | Laplace Mean ± Std | Gaussian Mean ± Std | Staircase Mean ± Std | RRonBins Mean ± Std | Ours Mean ± Std | Unbiased Mean ± Std |
|---|---|---|---|---|---|---|
| 0.05 | 14.2061 ± 12.1058 | 59.1688 ± 25.9966 | 9.9673 ± 3.7562 | 1.6680 ± 0.1241 | **1.5470 ± 0.1034** | 1.6307±0.1155 |
| 0.1 | 4.8926 ± 0.8567 | 14.6442 ± 10.7207 | 5.6695 ± 1.6705 | 1.6938 ± 0.1279 | **1.5400 ± 0.0894** | 1.6554±0.0542 |
| 0.3 | 2.2165 ± 0.5216 | 4.8875 ± 0.9392 | 2.6639 ± 1.3920 | 1.6132 ± 0.1017 | **1.5035 ± 0.0615** | 1.5656±0.0625 |
| 0.5 | **1.5666 ± 0.2492** | 3.4958 ± 0.9716 | **1.4574 ±0.1180** | 1.5477 ± 0.0754 | **1.4537 ± 0.0868** | 1.4666±0.0644 |
| 0.8 | **1.1604 ±0.1397** | 2.8569 ± 0.8632 | **1.0189 ± 0.0868** | 1.5397 ± 0.0915 | **1.1232 ± 0.0370** | 1.42420±0.0700 |
| 1.0 | **1.0121 ± 0.1150** | 2.3481 ± 0.6479 | **0.8862 ±0.0933** | 1.4407 ± 0.0969 | **1.0726 ± 0.0673** | 1.3641±0.0617 |
| 1.5 | **0.8262 ± 0.0761** | 1.6685 ± 0.3689 | **0.7905 ± 0.0552** | 1.2294 ± 0.0516 | **0.8797 ± 0.0324** | 1.1896±0.0447 |
| 2 | **0.7527 ± 0.0380** | 1.4513 ± 0.2093 | **0.7608 ± 0.0559** | 1.0917 ± 0.1835 | **0.7946 ± 0.0831** | 1.0882±0.0657 |
| 3 | **0.7279 ± 0.1016** | 1.0725 ± 0.1550 | **0.7141 ± 0.0329** | 0.7920 ± 0.0604 | **0.6732 ± 0.0698** | 0.9587±0.0766 |
| 4 | **0.6562 ± 0.0495** | 0.8827 ± 0.0689 | 0.7087 ± 0.0573 | 0.6755 ± 0.0477 | **0.6325 ± 0.0227** | 0.7659±0.0261 |
| 6 | **0.6153 ± 0.0234** | 0.7588 ± 0.0725 | 0.7113 ± 0.0402 | 0.6219 ± 0.0438 | **0.6106 ± 0.0320** | 0.6676±0.0334 |
| 8 | **0.6065 ± 0.0463** | 0.7270 ± 0.0408 | 0.7166 ± 0.0376 | **0.6195 ± 0.0466** | **0.5990 ± 0.0258** | 0.6282±0.0358 |
| $+\infty$ | **0.5922 ± 0.0244** | **0.5922 ± 0.0244** | **0.5922 ± 0.0244** | 0.5912 ± 0.0301 | **0.5852 ± 0.0224** | 0.6110 ±0.0292 |

The comparison results can be found in Table 3 on the California Housing dataset. Our method achieves the best performance across all privacy budgets. While the Laplace, Staircase and Unbiased mechanisms perform better on some privacy budgets, our method is consistent in all cases.

## 6 CONCLUSIONS

In this paper, we introduce a novel algorithm for label differential privacy in regression. Our approach, called RPWithPrior, leverages a known global prior distribution to ensure $\epsilon$-label differential privacy. In scenarios where the global prior distribution is unknown, we propose an alternative algorithm that employs a histogram to estimate the prior probability density function. This estimated function is then utilized as a prior in RPWithPrior. Theoretical analysis demonstrates that our proposed algorithms guarantee $\epsilon$-label differential privacy. Furthermore, we provide numerical experiments to evaluate the effectiveness of our proposed method.

This work has its own limitations: This work focuses on the setting of univariate regression, where the label $y$ is a scalar, and does not extend to multivariate regression or non-scalar labels. A natural pathway for such an extension would be to assume that the randomized mechanisms operate independently across dimensions. Under an independence assumption, the one-dimensional randomized mechanism developed in this paper could be applied independently to each dimension $y_i$ of the multivariate label. A thorough investigation of multivariate extensions, including the relaxation of the independence assumption, is a compelling direction for future research. In addition, we acknowledge that the current work does not establish the unbiasedness of our proposed estimator within the label DP framework. A formal investigation into these properties is an important direction for future research.

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

## A PRELIMINARY

Before delving deeper into the topic, it is important to establish the definitions of neighboring datasets, differential privacy and label differential privacy.

**Definition A.1** (Neighboring datasets (Dwork et al., 2006b)). Two datasets $D$ and $D'$ are called the neighboring datasets if they differ in at most one elements. That is, one is a proper subset of the other and the larger dataset contains just one additional element.

**Definition A.2** (Differential Privacy (Dwork et al., 2006b)). Let $\mathcal{A}$ be a randomized algorithm which takes a dataset as an input. For any $\epsilon, \delta \in \mathbb{R}_+$, $\mathcal{A}$ is said to be $(\epsilon, \delta)$-differentially private ($(\epsilon, \delta)$-DP) if $P[\mathcal{A}(\mathcal{D}) \in \mathcal{S}] \leq e^\epsilon P[\mathcal{A}(\mathcal{D}') \in \mathcal{S}] + \delta$ holds for any neighboring training sets $\mathcal{D}, \mathcal{D}'$ and any output sets $\mathcal{S}$ of $\mathcal{A}$. When $\delta = 0$, we call $\mathcal{A}$ is $\epsilon$-differential privacy ($\epsilon$-DP).

The definition of label differential privacy can be stated as follows:

**Definition A.3** (Label Differential Privacy (Chaudhuri and Hsu, 2011)). Let $\mathcal{A}$ be a randomized algorithm that takes a dataset as an input. For any $\epsilon, \delta \in \mathbb{R}_+$, $\mathcal{A}$ is said to be $(\epsilon, \delta)$-label differentially private ($(\epsilon, \delta)$-label DP) if the following condition holds for any training sets $\mathcal{D}, \mathcal{D}'$ that differ only in the label of a single example and any output sets $\mathcal{S}$ of $\mathcal{A}$: $P[\mathcal{A}(\mathcal{D}) \in \mathcal{S}] \leq e^\epsilon P[\mathcal{A}(\mathcal{D}') \in \mathcal{S}] + \delta$. When $\delta = 0$, we refer to $\mathcal{A}$ as $\epsilon$-label differential private ($\epsilon$-label DP).

## B PROOF OF LEMMA 3.3

**Proposition B.1.** *Let* $\mathcal{I} = [A_1, A_2]$ *and* $\gamma = 2\zeta + e^{-\epsilon}(A_2 - A_1)$. *Assume* $f_{\tilde{Y}|Y}(\tilde{y}, y) = v \geq 0, \forall y \in \mathcal{I}, \forall \tilde{y} \in \mathcal{N}_y$. *When* $y \in \mathcal{I}$, *we can make the following statement*

$$\int_{\mathcal{N}_{\mathcal{I}}} f_{\tilde{Y}|Y}(\tilde{y}, y) d\tilde{y} \geq \gamma \cdot v. \tag{B.1}$$

*Proof.* For any $y \in \mathcal{I}$, we have

$$\int_{\mathcal{N}_{\mathcal{I}}} f_{\tilde{Y}|Y}(\tilde{y}, y)d\tilde{y} = \int_{\mathcal{N}_y} f_{\tilde{Y}|Y}(\tilde{y}, y)d\tilde{y} + \sum_{k \neq 0} \int_{\mathcal{N}_{y+2k\zeta} \cap \mathcal{N}_{\mathcal{I}}} f_{\tilde{Y}|Y}(\tilde{y}, y)d\tilde{y}$$

$$\geq \int_{\mathcal{N}_y} f_{\tilde{Y}|Y}(\tilde{y}, y)d\tilde{y} + e^{-\epsilon} \sum_{k \neq 0} \int_{\mathcal{N}_{y+2k\zeta} \cap \mathcal{N}_{\mathcal{I}}} f_{\tilde{Y}|Y}(\tilde{y}, P_{\mathcal{I}}(y + 2k\zeta))d\tilde{y},$$

where we have used the fact that $e^{\epsilon} f_{\tilde{Y}|Y}(\tilde{y}, y) \geq f_{\tilde{Y}|Y}(\tilde{y}, P_{\mathcal{I}}(y + 2k\zeta))$ for all $\tilde{y} \in \mathcal{N}_{y+2k\zeta} \cap \mathcal{N}_{\mathcal{I}}$ and

$$P_{\mathcal{I}}(x) = \begin{cases} A_2, & x > A_2, \\ x, & x \in \mathcal{I}, \\ A_1, & x < A_1. \end{cases}$$

- When $y + 2k\zeta \in \mathcal{I}$ and $\tilde{y} \in \mathcal{N}_{y+2k\zeta} \cap \mathcal{N}_{\mathcal{I}}$, we have $P_{\mathcal{I}}(y + 2k\zeta) = y + 2k\zeta$, $\tilde{y} \in \mathcal{N}_{y+2k\zeta}$ and $|\tilde{y} - P_{\mathcal{I}}(y + 2k\zeta)| \leq \zeta$,

- when $y + 2k\zeta > A_2$ and $\tilde{y} \in \mathcal{N}_{y+2k\zeta} \cap \mathcal{N}_{\mathcal{I}}$, we have $y + (2k-1)\zeta \leq \tilde{y} \leq A_2 + \zeta$, $P_{\mathcal{I}}(y + 2k\zeta) = A_2$ and

$$\tilde{y} - P_{\mathcal{I}}(y + 2k\zeta) = \tilde{y} - A_2 \geq \tilde{y} - (y + 2k\zeta) \geq (y + (2k-1)\zeta) - (y + 2k\zeta)$$
$$= -\zeta,$$
$$\tilde{y} - P_{\mathcal{I}}(y + 2k\zeta) = \tilde{y} - A_2 \leq A_2 + \zeta - A_2 = \zeta,$$

- when $y + 2k\zeta < A_1$ and $\tilde{y} \in \mathcal{N}_{y+2k\zeta} \cap \mathcal{N}_{\mathcal{I}}$, we have $A_1 - \zeta \leq \tilde{y} \leq y + (2k+1)\zeta$, $P_{\mathcal{I}}(y + 2k\zeta) = A_1$ and

$$\tilde{y} - P_{\mathcal{I}}(y + 2k\zeta) = \tilde{y} - A_1 \leq \tilde{y} - (y + 2k\zeta) \leq (y + (2k+1)\zeta) - (y + 2k\zeta)$$
$$= \zeta,$$
$$\tilde{y} - P_{\mathcal{I}}(y + 2k\zeta) = \tilde{y} - A_1 \geq A_1 - \zeta - A_1 = -\zeta,$$

which implies $\tilde{y} \in \mathcal{N}_{P_{\mathcal{I}}(y+2k\zeta)}$ when $\tilde{y} \in \mathcal{N}_{y+2k\zeta} \cap \mathcal{N}_{\mathcal{I}}$. Note that $P_{\mathcal{I}}(y + 2k\zeta) \in \mathcal{I}$, we have $f_{\tilde{Y}|Y}(\tilde{y}, P_{\mathcal{I}}(y + 2k\zeta)) = v$, from the assumption $f_{\tilde{Y}|Y}(\tilde{y}, y) = v \geq 0, \forall y \in \mathcal{I}, \forall \tilde{y} \in \mathcal{N}_y$. Then

$$\int_{\mathcal{N}_{\mathcal{I}}} f_{\tilde{Y}|Y}(\tilde{y}, y)d\tilde{y} \geq [2\zeta + e^{-\epsilon}(A_2 - A_1)] \cdot v.$$

$\square$

*Proof of Lemma 3.3.* When $y \in \mathcal{I}$, Equation (B.1) holds from Proposition B.1. We can conclude that the first and second constraints of equation (3.3) imply that

$$1 \geq \int_{\mathcal{N}_{\mathcal{I}}} f_{\tilde{Y}|Y}(\tilde{y}, y)d\tilde{y} \geq \gamma \cdot v, \ \forall y \in \mathcal{I}.$$

which implies $v \leq 1/\gamma$. Then

$$\int_{\mathcal{I}} f_Y(y) \left[ \int_{\mathcal{N}_y} f_{\tilde{Y}|Y}(\tilde{y}, y)d\tilde{y} \right] dy = 2\zeta v \int_{\mathcal{I}} f_Y(y)dy \leq \frac{2\zeta}{\gamma} \int_{\mathcal{I}} f_Y(y)dy.$$

When $v = 1/\gamma$, the objective function achieve the maximal value. By the second constraint in (3.3), we have

$$f_{\tilde{Y}|Y}(\tilde{y}, y') \geq \frac{e^{-\epsilon}}{\gamma}, \forall \tilde{y} \in \mathcal{N}_{\mathcal{I}}, y' \in \mathcal{I}. \tag{B.2}$$

From (B.2) and $f_{\tilde{Y}|Y}(\tilde{y}, y') = 1/\gamma, \forall \tilde{y} \in \mathcal{N}_{y'}, y' \in \mathcal{I}$ when the objective function achieve the maximal value, we have

$$\frac{e^{-\epsilon}(A_2 - A_1)}{\gamma} \leq \int_{\mathcal{N}_{\mathcal{I}}/\mathcal{N}_{y'}} f_{\tilde{Y}|Y}(\tilde{y}, y')d\tilde{y} \leq 1 - \int_{\mathcal{N}_{y'}} f_{\tilde{Y}|Y}(\tilde{y}, y')d\tilde{y} = \frac{e^{-\epsilon}(A_2 - A_1)}{\gamma},$$

which implies

$$f_{\tilde{Y}|Y}(\tilde{y}, y') = \begin{cases} \frac{e^{-\epsilon}}{\gamma}, & \forall \tilde{y} \in \mathcal{N}_{\mathcal{I}}/\mathcal{N}_{y'}, y' \in \mathcal{I}, \\ 0, & \forall \tilde{y} \notin \mathcal{N}_{\mathcal{I}}. \end{cases} \text{ almost everywhere.}$$

We conclude the result. $\square$

## C   PROOF OF LEMMA 3.5

*Proof.* We get the derivatives about $A_1, A_2$, then

$$\frac{\partial F(A_1, A_2)}{\partial A_1}$$

$$=[-f_Y(A_1)]\frac{2\zeta}{2\zeta + e^{-\epsilon}(A_2 - A_1)}$$

$$+ \int_{A_1}^{A_2} f_Y(y)dy(2\zeta)(-1)(2\zeta + e^{-\epsilon}(A_2 - A_1))^{-2}(-1)e^{-\epsilon}$$

$$=\frac{-2\zeta}{[2\zeta + e^{-\epsilon}(A_2 - A_1)]^2}\left[f_Y(A_1)(2\zeta + e^{-\epsilon}(A_2 - A_1)) - e^{-\epsilon}\int_{A_1}^{A_2} f_Y(y)dy\right]$$

$$\triangleq \frac{-2\zeta}{[2\zeta + e^{-\epsilon}(A_2 - A_1)]^2}g_1(A_1, A_2),$$

and

$$\frac{\partial F(A_1, A_2)}{\partial A_2}$$

$$=\frac{2\zeta}{[2\zeta + e^{-\epsilon}(A_2 - A_1)]^2}\left[f_Y(A_2)(2\zeta + e^{-\epsilon}(A_2 - A_1)) - e^{-\epsilon}\int_{A_1}^{A_2} f_Y(y)dy\right]$$

$$\triangleq \frac{2\zeta}{[2\zeta + e^{-\epsilon}(A_2 - A_1)]^2}g_2(A_1, A_2),$$

where

$$g_1(A_1, A_2) = f_Y(A_1)(2\zeta + e^{-\epsilon}(A_2 - A_1)) - e^{-\epsilon}\int_{A_1}^{A_2} f_Y(y)dy,$$

$$g_2(A_1, A_2) = f_Y(A_2)(2\zeta + e^{-\epsilon}(A_2 - A_1)) - e^{-\epsilon}\int_{A_1}^{A_2} f_Y(y)dy.$$

When $A_1 < n_0$, $\frac{\partial F(A_1, A_2)}{\partial A_1} = \frac{2\zeta}{[2\zeta + e^{-\epsilon}(A_2 - A_1)]^2}e^{-\epsilon}\int_{A_1}^{A_2} f_Y(y)dy > 0$ for any $A_2 > A_1$. Then $F(A_1, A_2)$ is increasing when $A_1 \in (-\infty, n_0)$. Similarly, When $A_2 > n_k$, $\frac{\partial F(A_1, A_2)}{\partial A_2} = -\frac{2\zeta}{[2\zeta + e^{-\epsilon}(A_2 - A_1)]^2}e^{-\epsilon}\int_{A_1}^{A_2} f_Y(y)dy < 0$ for any $A_1 < A_2$. Then $F(A_1, A_2)$ is decreasing when $A_2 \in (n_k, \infty)$.

In the following, we consider $A_1, A_2 \in [n_0, n_k]$. Assume $A_1 \in [n_i, n_{i+1})$ and $A_2 \in [n_j, n_{j+1}]$ with $i \leq j$, then

$$g_1(A_1, A_2)$$

$$=\alpha_i(2\zeta + (A_2 - A_1)e^{-\epsilon}) - e^{-\epsilon}\left[\alpha_i(n_{i+1} - A_1) + \sum_{\ell=i+1}^{j-1} \alpha_\ell(n_{\ell+1} - n_\ell) + \alpha_j(A_2 - n_j)\right]$$

$$=e^{-\epsilon}(\alpha_i - \alpha_j)A_2 + c_1 = -(e^{-\epsilon}(\alpha_j - \alpha_i)A_2 - c_1),$$

$$g_2(A_1, A_2)$$

$$=\alpha_j(2\zeta + (A_2 - A_1)e^{-\epsilon}) - e^{-\epsilon}\left[\alpha_i(n_{i+1} - A_1) + \sum_{\ell=i+1}^{j-1} \alpha_\ell(n_{\ell+1} - n_\ell) + \alpha_j(A_2 - n_j)\right]$$

$$=e^{-\epsilon}(\alpha_i - \alpha_j)A_1 + c_2,$$

where $c_1 = 2\zeta\alpha_i - e^{-\epsilon}h$, $c_2 = 2\zeta\alpha_j - e^{-\epsilon}h$ and $h = \alpha_i n_{i+1} + \sum_{\ell=i+1}^{j-1} \alpha_\ell(n_{\ell+1} - n_\ell) - \alpha_j n_j$.

We also discuss the monotonicity of $F(A_1, A_2)$ as follows:

Define $d_{11} = e^{-\epsilon}(\alpha_j - \alpha_i)n_j - c_1$, $d_{12} = e^{-\epsilon}(\alpha_j - \alpha_i)n_{j+1} - c_1$.

- when $\max(d_{11}, d_{12}) < 0$, $F(A_1, A_2)$ is decreasing about $A_1 \in (n_i, n_{i+1})$ when $A_2 \in (n_j, n_{j+1})$,

- when $\min(d_{11}, d_{12}) > 0$, $F(A_1, A_2)$ is increasing about $A_1 \in (n_i, n_{i+1})$ when $A_2 \in (n_j, n_{j+1})$,

- when $d_{11}d_{12} < 0$, let $e_2 = \frac{c_1}{e^{-\epsilon}(\alpha_j - \alpha_i)}$, which must be in the interval $(n_j, n_{j+1})$.

  - When $d_{11} < 0$ and $d_{12} > 0$, $F(A_1, A_2)$ is decreasing about $A_1 \in (n_i, n_{i+1})$ for $A_2 \in (n_j, e_2)$ and $F(A_1, A_2)$ is increasing about $A_1 \in (n_i, n_{i+1})$ for $A_2 \in (e_2, n_{j+1})$.
  - When $d_{11} > 0$ and $d_{12} < 0$, $F(A_1, A_2)$ is increasing about $A_1 \in (n_i, n_{i+1})$ for $A_2 \in (n_j, e_2)$ and $F(A_1, A_2)$ is decreasing about $A_1 \in (n_i, n_{i+1})$ for $A_2 \in (e_2, n_{j+1})$.

Define $d_{21} = e^{-\epsilon}(\alpha_i - \alpha_j)n_i + c_2$ and $d_{22} = e^{-\epsilon}(\alpha_i - \alpha_j)n_{i+1} + c_2$. Therefore,

- when $\max(d_{21}, d_{22}) < 0$, $F(A_1, A_2)$ is decreasing about $A_2 \in (n_j, n_{j+1})$ when $A_1 \in (n_i, n_{i+1})$,

- when $\min(d_{21}, d_{22}) > 0$, $F(A_1, A_2)$ is increasing about $A_2 \in (n_j, n_{j+1})$ when $A_1 \in (n_i, n_{i+1})$,

- when $d_{21}d_{22} < 0$, let $e_1 = \frac{c_2}{e^{-\epsilon}(\alpha_j - \alpha_i)}$, which must be in the interval $(n_i, n_{i+1})$.

  - When $d_{21} < 0$ and $d_{22} > 0$, $F(A_1, A_2)$ is decreasing about $A_2 \in (n_j, n_{j+1})$ for $A_1 \in (n_i, e_1)$ and $F(A_1, A_2)$ is increasing about $A_2 \in (n_j, n_{j+1})$ for $A_1 \in (e_1, n_{i+1})$.
  - When $d_{21} > 0$ and $d_{22} < 0$, $F(A_1, A_2)$ is increasing about $A_2 \in (n_j, n_{j+1})$ for $A_1 \in (n_i, e_1)$ and $F(A_1, A_2)$ is decreasing about $A_2 \in (n_j, n_{j+1})$ for $A_1 \in (e_1, n_{i+1})$.

$\square$

## D   RPWITHPRIOR$_\epsilon$ IS $\epsilon$-LABEL DP

*Proof.* Note that $f_{\tilde{Y}|Y}(\tilde{y}, y)$ is represented by (3.4) when $y \in \mathcal{I}$, otherwise $f_{\tilde{Y}|Y}(\tilde{y}, y)$ satisfies (3.5). For all $y$, we have $f_{\tilde{Y}|Y}(\tilde{y}, y) = 0$, when $\tilde{y} \notin \mathcal{N}_\mathcal{I}$. This means the randomized response $\tilde{y} \in \mathcal{N}_\mathcal{I}$ whenever the original response is.

For any $\tilde{y} \in \mathcal{N}_\mathcal{I}$ and any $y$, $\max(f_{\tilde{Y}|Y}(\tilde{y}, y)) = \frac{1}{\gamma}$ and $\min(f_{\tilde{Y}|Y}(\tilde{y}, y)) = \frac{e^{-\epsilon}}{\gamma}$, which satisfies the $\epsilon$-label DP property. Therefore, we can conclude the result. $\square$

## E   ALGORITHM 3 IS $(\epsilon_1 + \epsilon_2)$-LABEL DP.

*Proof.* Algorithm 3 splits the privacy budget into $\epsilon_1, \epsilon_2$, which follows that the entire algorithm is $(\epsilon_1 + \epsilon_2)$-label DP. $\square$

## F   ERROR ANALYSIS

**Theorem F.1** (Theorem 1 in (Siegel, 2023)). *Let $\Omega = [0, 1]^d$ be the unit cube in $\mathbb{R}^d$ and let $0 < s < +\infty$ and $1 \le p, q \le +\infty$. Assume that $\frac{1}{q} - \frac{1}{p} < \frac{s}{d}$, which guarantees that we have the Sobolev space $W^s(L_q(\Omega))$ is a compact embedding*

$$W^s(L_q(\Omega)) \subset\subset L^p(\Omega).$$

*Then we have that*

$$\inf_{f_L \in \Upsilon^{25d+31, L}(\mathbb{R}^d)} \|f - f_L\|_{L_p(\Omega)} \le C \|f\|_{W^s(L_q(\Omega))} L^{-2s/d}$$

*for a constant $C := C(s, q, p, d) < +\infty$.*

**Theorem F.2** (Theorem 2 in (Siegel, 2023)). *Let $\Omega = [0,1]^d$ be the unit cube in $\mathbb{R}^d$ and let $0 < s < +\infty$ and $1 \le p, q \le +\infty$. Assume that $\frac{1}{q} - \frac{1}{p} < \frac{s}{d}$, which guarantees that we have the Besov space $B_r^s(L_q(\Omega))$ is a compact embedding*

$$B_r^s(L_q(\Omega)) \subset\subset L^p(\Omega).$$

*Then we have that*

$$\inf_{f_L \in \Upsilon^{25d+31,L}(\mathbb{R}^d)} \|f - f_L\|_{L_p(\Omega)} \le C\|f\|_{B_r^s(L_q(\Omega))} L^{-2s/d}$$

*for a constant $C := C(s,q,p,d) < +\infty$.*

**Theorem F.3.** *Let $\ell(\cdot,\cdot)$ be a loss function, $x, y, \tilde{y}$ be a predictor, a true response and a randomized response, respectively. Let $\tilde{g}(x)$ be a predicted regression function by $(x, \tilde{y})$. When the assumptions satisfy in Theorems F.1 and F.2 and $\tilde{g}(x) \in \Upsilon^{25d+31,L}(\mathbb{R}^d), \forall L \in \mathbb{Z}^+$, where $\Upsilon^{25d+31,L}(\mathbb{R}^d)$ is a class of neural network with width $25d + 31$ and depth $L$. Then the error can be bounded as*

$$\mathbb{E}_{x,y,\tilde{y}}[\ell(\tilde{g}(x),y)] \le 2C\|f\|_{W^s(L_q(\Omega))} L^{-2s/d} + \frac{4}{\gamma}\zeta P_{\mathcal{I}}(y) + \frac{e^{-\epsilon}}{2\gamma}(A_1 + A_2)(A_2 - A_1 + 2\zeta),$$

$$\mathbb{E}_{x,y,\tilde{y}}[\ell(\tilde{g}(x),y)] \le 2C\|f\|_{B_r^s(L_q(\Omega))} L^{-2s/d} + \frac{4}{\gamma}\zeta P_{\mathcal{I}}(y) + \frac{e^{-\epsilon}}{2\gamma}(A_1 + A_2)(A_2 - A_1 + 2\zeta),$$

*for a constant $C := C(s,q,p,d) < +\infty$, where*

$$P_{\mathcal{I}}(y) = \begin{cases} A_2, & y > A_2, \\ y, & y \in \mathcal{I}, \\ A_1, & y < A_1. \end{cases}$$

*Proof.* Let $\Omega = [0,1]^d$ and $\ell(\cdot,\cdot)$ be $L_p$ norm, $x, y, \tilde{y}$ be a predictor, a true response and a randomized response, respectively. Let $\tilde{g}(x)$ be a predicted regression function by $(x, \tilde{y})$ and $g(x) \in L_p(\Omega)$ be a true regression function such that $g(x) = \tilde{y}$. Then the error can be bounded as

$$\mathbb{E}_{x,y,\tilde{y}}[\ell(\tilde{g}(x),y)] \le 2\mathbb{E}_{x,y,\tilde{y}}[\ell(\tilde{g}(x),\tilde{y})] + 2\mathbb{E}_{x,y,\tilde{y}}[\ell(\tilde{y},y)] := 2E_1 + 2E_2.$$

In deep learning, $E_1$ (model risk) decays as network complexity increases. From Siegel (2023), we have $E_1 \le C\|f\|_{W^s(L_q(\Omega))} L^{-2s/d}$ or $E_1 \le C\|f\|_{B_r^s(L_q(\Omega))} L^{-2s/d}$ for a constant $C := C(s,q,p,d) < +\infty$.

Note that $E_2 = \mathbb{E}_{x,y,\tilde{y}}[\ell(\tilde{y},y)] = \mathbb{E}_{y,\tilde{y}}[\ell(\tilde{y},y)] = \mathbb{E}_y[\mathbb{E}_{\tilde{y}|y}\ell(\tilde{y},y)]$ and

$$f_{\tilde{Y}|Y}(\tilde{y},y) = \begin{cases} 1/\gamma, & \text{if } \tilde{y} \in \mathcal{N}_{P_{\mathcal{I}}(y)}, \\ e^{-\epsilon}/\gamma, & \text{if } \tilde{y} \in \mathcal{N}_{\mathcal{I}}/\mathcal{N}_{P_{\mathcal{I}}(y)}, \\ 0, & \text{otherwise}, \end{cases} \quad P_{\mathcal{I}}(y) = \begin{cases} A_2, & y > A_2, \\ y, & y \in \mathcal{I}, \\ A_1, & y < A_1, \end{cases}$$

then

$$\mathbb{E}_{\tilde{y}|y}\ell(\tilde{y},y) = \int_{\mathcal{N}_{\mathcal{I}}} \frac{1}{2}(y - \tilde{y})^2 f_{\tilde{Y}|Y}(\tilde{y},y) d\tilde{y}$$

$$= \int_{\mathcal{N}_{\mathcal{I}}/\mathcal{N}_{P_{\mathcal{I}}(y)}} \frac{1}{2}(y - \tilde{y})^2 f_{\tilde{Y}|Y}(\tilde{y},y) d\tilde{y} + \int_{\mathcal{N}_{P_{\mathcal{I}}(y)}} \frac{1}{2}(y - \tilde{y})^2 f_{\tilde{Y}|Y}(\tilde{y},y) d\tilde{y}$$

$$= \frac{2}{\gamma}\zeta P_{\mathcal{I}}(y) + \frac{e^{-\epsilon}}{2\gamma}(A_1 + A_2)(A_2 - A_1 + 2\zeta).$$

Hence $E_2 = \mathbb{E}_y[\frac{2}{\gamma}\zeta P_{\mathcal{I}}(y) + \frac{e^{-\epsilon}}{2\gamma}(A_1 + A_2)(A_2 - A_1 + 2\zeta)] = \frac{2}{\gamma}\zeta\mathbb{E}_y[P_{\mathcal{I}}(y)] + \frac{e^{-\epsilon}}{2\gamma}(A_1 + A_2)(A_2 - A_1 + 2\zeta)$. In summary, we can conclude the result. $\square$

## G THE HISTOGRAM ALGORITHM

Let $\mathcal{M}_{\epsilon,\text{Lap}}(y)$ denote the randomized response of $y$ under the Laplace mechanism with privacy budget $\epsilon$. The histogram-based algorithm for prior estimation is then formulated as follows:

---

**Algorithm 4** Estimate the prior by histogram ($\text{Hist}_\epsilon$)

---

**Require:** Dataset $D$, a positive parameter $\sigma > 0$.

1: $\mu = \frac{1}{\#(D)} \sum_{(x,y) \in D} \mathcal{M}_{\epsilon,\text{Lap}}(y)$,          $\triangleright$ Sample mean

2: Compute integers $k_0$ and $k_1$ satisfying

$$\mu + k_0\sigma \leq \min_{(x,y) \in D} \mathcal{M}_{\epsilon,\text{Lap}}(y) < \mu + (k_0+1)\sigma, \ \mu + k_1\sigma < \max_{(x,y) \in D} \mathcal{M}_{\epsilon,\text{Lap}}(y) \leq \mu + (k_1+1)\sigma.$$

3: Compute nodes of histogram

$$n_i = \begin{cases} \min_{(x,y) \in D} \mathcal{M}_{\epsilon,\text{Lap}}(y), & \text{if } i = 0, \\ \mu + (i + k_0)\sigma, & \text{if } i = 1, \cdots, k_1 - k_0, \\ \max_{(x,y) \in D} \mathcal{M}_{\epsilon,\text{Lap}}(y), & \text{if } i = k_1 - k_0 + 1. \end{cases}$$

4: Compute intervals of histogram

$$J_k = \begin{cases} [n_k, n_{k+1}), & k \in [k_1 - k_0 - 1], \\ [n_k, n_{k+1}], & k = k_1 - k_0, \end{cases} \quad S_k = \{y | \mathcal{M}_{\epsilon,\text{Lap}}(y) \in J_k, (x,y) \in D\}.$$

5: $\alpha_k = \#(S_k)/\#(D), k = 0, \cdots, k_1 - k_0$.          $\triangleright$ Histogram expression

6: **return** $\{(J_k, \alpha_k)\}$.

---

# H DETAILS OF NUMERICAL EXPERIMENTS

In numerical experiment section, we conduct a comprehensive performance evaluation of our proposed method and compare it with several existing mechanisms, including the Gaussian mechanism (Dwork et al., 2006a), the Laplace mechanism (Dwork et al., 2006b), the staircase mechanism (Geng and Viswanath, 2014), and RRonBins (Ghazi et al., 2022). To avoid overfitting, we incorporate an $L_2$ regularizer into the loss functions of all these mechanisms. It's worth noting that the Gaussian mechanism is an approximate differential privacy method, and as such, we cannot set $\zeta = 0$ (as defined in the DP framework) in our numerical tests. Here, we set $\zeta = 10^{-4}$ in the following experiments. The training procedure for all datasets was conducted without early stopping. All experiments were conducted on a workstation equipped with an Intel Xeon W-2245 CPU, NVIDIA RTX 3080 Ti GPU, and 128GB RAM, using PyTorch under CUDA 11.7.

## H.1 THE COMMUNITIES AND CRIME DATASET

The Communities and Crime dataset is a combined socio-economic data from the 1990 US Census, law enforcement data from the 1990 US LEMAS survey, and crime data from the 1995 FBI UCR, which can be downloaded from the website of UCI machine learning repository[1]. The creator is Michael Redmond from La Salle University. It is multivariate with 1994 instances and 128 attributes. Note that there are some missing values in the dataset, it is necessary to deal with in advance. In this paper, we use the similar technique as the paper of Selective Regression Under Fairness Criteria (Shah et al., 2022). First of all, we remove the useless attributes for regression, including *state, county, community, communityname, fold*. For these attributes with missing values, we remove all samples with missing values unless the attribute is *OtherPerCap*, whose missing value is replaced by the mean of all the values in this attribute (Chi et al., 2021). After preprocessing, there are 101 attributes left.

Next, we investigate the relationship between the response variable, specifically the *ViolentCrimesPerPop*, and the remaining features. To accomplish this, we employ a simple neural network consisting of three fully-connected layers. The first and second layers of the neural network are fully-connected layers followed by Rectified Linear Unit (ReLU) activation function. The third layer is also a fully-connected layer.

We train it using the Adam optimizer with an initial learning rate of 0.001. The model undergoes training for a total of 50 epochs and set the batchsize equal to 256, with the learning rate decayed by a factor of 10 at the 25th epoch. For the $L_2$ regularizer, we set the factor to weight_decay = 1e-4.

---

[1]https://archive.ics.uci.edu/ml/datasets/communities+and+crime

## H.2 THE CRITEO SPONSORED SEARCH CONVERSION LOG DATASET

The Criteo Sponsored Search Conversion Log Dataset is public and can be downloaded from the Criteo AI Lab website [2]. This dataset represents a sample of 90 days of Criteo live traffic data. Each line corresponds to one click (product related advertisement) that was displayed to a user. For each advertisement, we have detailed information about the product. Further, it also provides information on whether the click led to a conversion, amount of conversion and the time between the click and the conversion. Data has been sub-sampled and anonymized so as not to disclose proprietary elements.

In the Criteo Sponsored Search Conversion Log Dataset, there are 23 attributes (the first three are outcome/responses and the remaining are features). We use 21 of them except *Sale, click_timestamp*, where *SalesAmountInEuro* is considered as the response and the remaining are features. In the features, *Time_delay_for_conversion, nb_clicks_1week* are integers, *product_price* is float, and *product_age_group, device_type, audience_id, product_gender, product_brand, product_category (1-7), product_country, product_id, product_title, partner_id, user_id* are hashed features for user privacy. Therefore, it is necessary to do preprocessing of the hashed features before being used for response DP. We use the very similar technique as that for kaggle-display-Advertising-challenge-dataset (see the webpage `https://github.com/rixwew/pytorch-fm/blob/master/torchfm/dataset/criteo.py`) with a small modification. We count all the unique values for each feature and map them to their corresponding unique value if the occurrence is large than the preassigned thresholding, otherwise consider as a single out-of-vocabulary. For the architecture of the neural networks used for regression, we benefit from MentorNet (Jiang et al., 2018) used for the non-numerical features (all but attributes *Time_delay_for_conversion, nb_clicks_1week, product_price*) and put them to embeddings. Then we combine the embedded features with *Time_delay_for_conversion, nb_clicks_1week, product_price* together as the input and feed them into a neural network with 3 fully-connected layers.

Now, we are ready for response DP. We plan to predict the relationship between response (*SalesAmountInEuro*) and the features. There are 15,995,634 samples in the dataset, but it is unfortunate that some of them are with no conversion happened (corresponding to *SalesAmountInEuro*$= -1$) and some are too large (the largest one is 62,458.773). We have 1,645,977 samples left after removing those useless ones (-1 or larger than 400). We set the maximum number of epochs to 50 and the batch size to 8192. The model is trained using the RMSProp optimizer with an initial learning rate of 1e-3, and the learning rate is decayed using the *CosineAnnealingLR* method. Additionally, we set the $L_2$ regularizer parameter weight_decay to 1e-4.

## H.3 THE CALIFORNIA HOUSING DATASET

This California Housing dataset was derived from the 1990 U.S. census, which can be obtained from the StatLib repository[3] or import through the command *fetch_california_housing*[4] in sklearn.datasets.

Table 4: Attribute Information.

| Name | Description |
|------|-------------|
| MedInc | median income in block group |
| HouseAge | median house age in block group |
| AveRooms | average number of rooms per household |
| AveBedrms | average number of bedrooms per household |
| Population | block group population |
| AveOccup | average number of household members |
| Latitude | block group latitude |
| Longitude | block group longitude |

---

[2] `https://ailab.criteo.com/criteo-sponsored-search-conversion-log-dataset/`
[3] `https://www.dcc.fc.up.pt/~ltorgo/Regression/cal_housing.html`
[4] `https://scikit-learn.org/stable/modules/generated/sklearn.datasets.fetch_california_housing.html#sklearn.datasets.fetch_california_housing`

In this dataset, we have information regarding the demography (income, population, house occupancy) in the districts, the location of the districts (latitude, longitude), and general information regarding the house in the districts (number of rooms, number of bedrooms, age of the house). There are 20640 instances, 9 attributes: 8 numeric, predictive attributes and the target, where the attribute information is shown in Table 4. Since these statistics are at the granularity of the district, they correspond to averages or medians. There is no missing attribute values, we use the data directly.

For the architecture of the regression neural networks, we employ a 3-layer fully-connected neural network. The model is trained for a total of 50 epochs, with a batch size set to 256. During training, we utilize the Adam optimizer with an initial learning rate of $0.001$. The learning rate is decayed by a factor of 10 at the 25th epoch. For $L_2$ regularization, we set the regularization factor to weight_decay = 1e-4.

## H.4 PARAMETERS

In the following, we will report the parameters used in the Communities and Crime Dataset, the Criteo Sponsored Search Conversion Log Dataset and the California Housing Dataset, where $\epsilon = \epsilon_1 + \epsilon_2$. We choose parameters manually, similar as grid search. From the above table, we find it is not

Table 5: Parameters.

| | Crime | | | Criteo | | | Housing | |
|---|---|---|---|---|---|---|---|---|
| $\epsilon$ | $\epsilon_1$ | $\zeta$ | $\epsilon$ | $\epsilon_1$ | $\zeta$ | $\epsilon$ | $\epsilon_1$ | $\zeta$ |
| 0.05 | $\epsilon/2$ | 0.4 | 0.05 | 0.017 | 70 | 0.05 | 0.017 | 0.7 |
| 0.1 | $\epsilon/2$ | 0.4 | 0.1 | 0.017 | 70 | 0.1 | 0.017 | 0.5 |
| 0.3 | $\epsilon/2$ | 0.4 | 0.3 | 0.017 | 70 | 0.3 | 0.017 | 1.2 |
| 0.5 | $\epsilon/2$ | 0.4 | 0.5 | 0.017 | 70 | 0.5 | 0.017 | 1 |
| 0.8 | $\epsilon/2$ | 0.4 | 0.8 | 0.017 | 70 | 0.8 | 0.01 | 2.2 |
| 1 | $\epsilon/2$ | 0.4 | 1 | 0.017 | 70 | 1 | 0.008 | 1.5 |
| 1.5 | $\epsilon/2.5$ | 0.4 | 1.5 | 0.01 | 70 | 1.5 | 0.008 | 1.5 |
| 2 | $\epsilon/2.5$ | 0.4 | 2 | 0.01 | 70 | 2 | 0.008 | 1.5 |
| 3 | $\epsilon/3$ | 0.4 | 3 | 0.01 | 70 | 3 | 0.007 | 1.4 |
| 4 | $\epsilon/3.5$ | 0.4 | 4 | 0.005 | 70 | 4 | 0.007 | 1.2 |
| 6 | $\epsilon/6$ | 0.4 | 6 | 0.003 | 70 | 6 | 0.007 | 0.7 |
| 8 | $\epsilon/7$ | 0.4 | 8 | 0.002 | 70 | 8 | 0.007 | 0.1 |
| $+\infty$ | $+\infty$ | 0.1 | $+\infty$ | $+\infty$ | 0.8 | $+\infty$ | $+\infty$ | 0.1 |

sensitive of the parameter $\zeta$.

In addition, the performance of our method is largely insensitive to the value of $\sigma$. In empirical tests, varying $\sigma$ by a factor of $\alpha$ around the fixed values $0.5, 1, 2$ resulted in negligible change to the outcome. This robustness is expected, as Algorithm 4 is used solely to estimate the prior distribution for determining the optimal interval.

## H.5 COMPUTATIONAL COMPLEXITY

To save storage space and reduce computational costs, the authors of RRonBins discretized the labels by rounding down the original responses to integer values (referred to as discretized samples). For the remaining methods, the original responses are retained.

In Table 6, we provide the runtime and unique number of samples in the training process. Our method is computationally efficient, comparable to additive noise mechanisms and significantly outperforming the RRonBins mechanism.

## I SIMULATED DATASETS

We do numerical experiments on simulated dataset (both linear and nonlinear) as follows: let $X \in \mathbb{R}^{n \times d}$, $y \in \mathbb{R}^n$, where $n$ is the sample size and $d$ is the dimension. In the both cases, we generate the feature matrix $X$ randomly obeying Standard Normal distribution.

Table 6: Runtime and Unique number ($n$) of samples or discretized samples in the training process.

| Method | Crime | | Criteo | | Housing | |
|---|---|---|---|---|---|---|
| | Time | $n$ | Time | $n$ | Time | $n$ |
| Laplace | 0.288116455 | 1595 | **3.408373356** | 1316781 | **0.279757977** | 16512 |
| Gaussian | 0.285983801 | 1595 | 3.42813 | 1316781 | 0.291029215 | 16512 |
| Staircase | 0.285481691 | 1595 | 3.538956881 | 1316781 | 0.312218666 | 16512 |
| RRonBins | 0.567934513 | 94 | 8.24254632 | 401 | 6.187591076 | 472 |
| Ours | **0.279293299** | 1595 | 3.44035 | 1316781 | 0.297134399 | 16512 |

- **Linear**: we also generate the vector $A \in \mathbb{R}^d$ randomly obeying Standard Normal distribution and set $y = \sum_{i=1}^{d} A[i]X[:,i]$, where $X[:,i]$ is the $i$-th column of $X$. The numerical results are as follows: From the above table, we conclude all the methods can get better results for

Table 7: Comparison results on the linear simulated dataset.

| Privacy Budget | Laplace | Gaussian | Staircase | RRonBins | Ours | Unbiased |
|---|---|---|---|---|---|---|
| | Mean ± Std | Mean ± Std | Mean ± Std | Mean ± Std | Mean ± Std | Mean ± Std |
| 0.05 | 6.7619±3.0561 | 10.8907±3.8553 | 12.1042±3.3601 | 10.1382±4.3226 | 9.2342±4.7130 | 11.8071±5.0206 |
| 0.1 | 11.6518±6.5433 | 11.2220±3.1782 | 9.9774±5.1210 | 10.2426±3.7583 | 5.7169±1.5244 | 10.4242±4.9875 |
| 0.3 | 10.5618±4.6079 | 10.7778±3.5415 | 7.7710±3.4123 | 8.7568±3.2488 | 10.3128±5.899 | 9.9343±4.0548 |
| 0.5 | 8.3280±5.66974 | 9.1884±5.1005 | 9.1060±3.0008 | 10.5588±3.4701 | 6.7450±3.6947 | 10.5926±3.7888 |
| 0.8 | 7.2296±3.2733 | 7.4162±3.9262 | 4.6615±2.4509 | 7.4461±3.7534 | 8.7830±3.8595 | 9.3589±4.1997 |
| 1 | 6.4560±2.9871 | 10.2868±3.1686 | 6.1859±4.6726 | 9.3996±1.7144 | 7.2708±2.5645 | 7.2313±3.9793 |
| 1.5 | 5.6707±3.4424 | 7.8765±3.3130 | 3.9385±2.9584 | 5.7334±2.2097 | 5.2094±2.2437 | 7.3531±3.1218 |
| 2 | 7.9935±4.1851 | 9.5327±4.7792 | 6.2754±3.5710 | 5.7551±2.4432 | 4.8456±3.1561 | 5.1702±2.2089 |
| 3 | 4.0967±1.2403 | 7.0141±5.1296 | 5.9760±3.2838 | 5.1670±1.5725 | 5.1234±1.3356 | 8.6822±5.4475 |
| 4 | 4.6560±2.1379 | 4.4751±2.5818 | 5.0915±3.5990 | 5.5471±2.3634 | 3.7567±3.0159 | 4.7316±2.0718 |
| 6 | 2.7771±1.2930 | 6.3369±4.1048 | 4.5594±2.9253 | 5.2050±3.7011 | 3.6992±2.9984 | 4.1146±2.7303 |
| 8 | 3.3923±2.1478 | 5.2665±3.3694 | 3.9599±1.8278 | 5.1476±3.1585 | 4.1412±1.8096 | 4.6723±2.8677 |
| ∞ | 0.0120±0.0050 | 0.0120±0.0050 | 0.0120±0.0050 | 0.0173±0.0053 | 0.0125±0.0069 | 0.0219±0.0086 |

linear simulated dataset.

- **Nonlinear**: $y = \sum_{i=1}^{d} \exp(X[:,i])$, where $X[:,i]$ is the $i$-th column of $X$. The numerical results are as follows:

Table 8: Comparison results on nonlinear simulated dataset.

| Privacy Budget | Laplace | Gaussian | Staircase | RRonBins | Ours | Unbiased |
|---|---|---|---|---|---|---|
| | Mean ± Std | Mean ± Std | Mean ± Std | Mean ± Std | Mean ± Std | Mean ± Std |
| 0.05 | 12187.3481±13439.9494 | 39383.0625±22130.0212 | 7379.2940±3660.319 | **312.2784±115.1406** | 155.4341±76.1839 | 1252.6487±537.9432 |
| 0.1 | 3565.2386±2162.5579 | 8309.4118±4890.6477 | 1871.4528±817.2671 | 451.3280±184.7202 | **53.9678±22.0496** | 796.6692±384.3283 |
| 0.3 | 392.2554±182.3960 | 3090.6682±1624.5669 | 513.4610±383.4003 | 348.5720±131.7954 | **42.0532±3.4690** | 2456.8288±3920.4370 |
| 0.5 | 162.0974±76.9350 | 1684.8579±1107.2722 | 132.1034±76.9102 | 340.9608±138.6306 | **40.4100±4.0826** | 563.4305±201.8678 |
| 0.8 | 79.2950±26.7152 | 426.5728±246.2399 | **53.2645±23.1531** | 344.2565±189.8461 | 37.4296±3.8043 | 731.2202±430.8711 |
| 1 | **61.0212±42.8874** | 289.4453±116.5660 | 48.6851±28.3232 | 236.5660±76.9058 | 17.0438±1.7695 | 665.3807±247.5771 |
| 1.5 | **40.5246±24.7389** | 200.5178±195.2500 | **37.3397±28.0381** | 352.5059±168.2677 | 26.2684±4.4789 | 414.7472±169.2534 |
| 2 | **25.0478±6.4657** | 83.4924±30.2726 | **18.0702±3.9350** | 221.9705±75.8004 | 21.3296±2.0643 | 396.3458±246.6874 |
| 3 | **15.2806±3.3548** | 56.2989±38.6322 | **16.1297±2.7485** | 91.2669±24.0746 | 16.5606±1.4494 | 156.4980±71.7431 |
| 4 | **13.1973±1.4805** | 34.6206±10.0915 | 17.7200±2.1758 | 60.4433±20.9969 | 15.4449±2.0659 | 105.0789±38.5969 |
| 6 | **11.9794±2.1062** | 20.6233±5.2040 | 19.2643±4.1972 | 25.6626±10.1864 | 12.587±5.8092 | 34.0717±22.5132 |
| 8 | **9.4974±3.0906** | 17.3384±3.6793 | 19.6195±3.9102 | **11.8751±2.8385** | 12.6356±2.6500 | **15.9530±3.6365** |
| ∞ | **6.4681±1.9020** | **6.4681±1.9020** | **6.4681±1.9020** | **5.9318±1.5030** | 6.6823±1.6788 | 7.2015±1.3027 |

From Table 8, our proposed method is much more consistent.

## J COMPARISONS BETWEEN SOME EXISTING METHODS FOR REGRESSION AND OURS

The methods of Ghazi et al. (2022) and Badanidiyuru et al. (2023) fundamentally convert a regression problem into a classification problem within the framework of Label Differential Privacy. In contrast, our method directly addresses the regression problem under Label DP.

In machine learning theory, regression and classification are distinct tasks, defined by their output spaces:

- Regression addresses estimation problems where the output space $Y$ is **continuous** and ordered, with the goal of predicting a real-valued quantity.
- Classification deals with problems where the output space is a **finite** set of **discrete** categories.

For RRonbins, this conversion is achieved through a multi-step process:

- **Rounding**: The original continuous responses (e.g., the Criteo Search dataset) are first rounded to integers.
- **Discretization (Binning)**: A dynamic programming algorithm is used to partition the rounded values into a finite set of bins, effectively creating a set of discrete classes.
- **Randomization**: A standard Randomized Response (RR) mechanism is applied over this finite set of bins.

For Badanidiyuru et al. (2023), the conversion is as follows:

- Unbiased Randomized rounding to finite points
- Construct a discrete randomized response space as follows:

$$L = ((e^\epsilon + |\mathcal{Y}| - 1)\min(\mathcal{Y}) - \sum_{y \in \mathcal{Y}} y)/(e^\epsilon - 1)$$

$$U = ((e^\epsilon + |\mathcal{Y}| - 1)\max(\mathcal{Y}) - \sum_{y \in \mathcal{Y}} y)/(e^\epsilon - 1)$$

$$\Delta = (U - L)/(n - 1)$$

where $\mathcal{Y}$ is the discretized space of the original responses. The randomized response space is

$$\hat{\mathcal{Y}} = (L, L + \Delta, \cdots, U - \Delta, U)$$

which contains $n$ discrete points.

- Compute the transition matrix from the original responses to the randomized responses and randomization.

Our method solve the regression problem directly.

- **No Discretization**: Our methodology is fundamentally continuous. The original responses are treated as continuous random variables and are never discretized at any stage of the process.
- **Theoretical Support**: The primary contribution of our theoretical results is to establish the foundation for this continuous approach. They rigorously support an algorithm designed to map a continuous random variable to another continuous random variable while satisfying the required privacy constraints.

## J.1 THEORETICAL COMPARISON WITH THE ALGORITHM BY BADANIDIYURU ET AL. (2023)

**Theorem J.1.** *We consider a nonnegative response space and assume $\frac{\sum_{\hat{y} \in \hat{\mathcal{Y}}} \hat{y} - |\hat{\mathcal{Y}}|}{e^\epsilon - 1} > (M - m)$, where $\hat{\mathcal{Y}}$ is the unbiased randomized space (see the outline of the algorithm by Badanidiyuru et al. (2023)), $|\hat{\mathcal{Y}}|$ is the cardinality of $\hat{\mathcal{Y}}$ and $m, M$ are the minimum and maximum values of $\hat{\mathcal{Y}}$. Let $\Omega = [0, 1]^d$ and $\ell(\cdot, \cdot)$ be $L_p$ norm, $x, y, \tilde{y}$ be a predictor, a true response and a randomized response, respectively. Let $\tilde{g}(x)$ be a predicted regression function by $(x, \tilde{y})$ and $g(x) \in L_p(\Omega)$ be a true regression function such that $g(x) = \tilde{y}$. Then the error can be bounded as*

$$\mathbb{E}_{x,y,\tilde{y}}[\ell(\tilde{g}(x), y)] \le 2C\|f\|L^{-2s/d} + \left(\frac{\sum_{\hat{y} \in \hat{\mathcal{Y}}} \hat{y} - |\hat{\mathcal{Y}}|}{e^\epsilon - 1}\right)^2,$$

*where $\|f\| = \|f\|_{W^s(L_q(\Omega))}$ or $\|f\| = \|f\|_{B_r^s(L_q(\Omega))}$.*

*Proof.* Note that

$$\mathbb{E}_{x,y,\tilde{y}}[\ell(\tilde{g}(x), y)] \leq 2\mathbb{E}_{x,y,\tilde{y}}[\ell(\tilde{g}(x), \tilde{y})] + 2\mathbb{E}_{x,y,\tilde{y}}[\ell(\tilde{y}, y)] := 2E_1 + 2E_2.$$

In deep learning, $E_1$ (model risk) decays as network complexity increases. From (Badanidiyuru et al., 2023), we have

$$E_1 \leq C\|f\|L^{-2s/d}$$

for a constant $C := C(s, q, p, d) < +\infty$.

Next, we estimate the upper and lower bounds of $E_2$. Since $\frac{\sum_{\hat{y} \in \hat{\mathcal{Y}}} \hat{y} - |\hat{\mathcal{Y}}|}{e^\epsilon - 1} > (M - m)$, we have $M - \frac{\sum_{\hat{y} \in \hat{\mathcal{Y}}} \hat{y} - |\hat{\mathcal{Y}}|}{e^\epsilon - 1} < m$. Note that $\mathcal{Y}$ and $\hat{\mathcal{Y}}$ share the same minimum and maximum value. That is, the minimum of the original response space $\hat{\mathcal{Y}}$ is larger than the maximum of the randomized response space $\tilde{\mathcal{Y}}$. Therefore,

$$\left(\frac{\sum_{\hat{y} \in \hat{\mathcal{Y}}} \hat{y} - |\hat{\mathcal{Y}}|}{e^\epsilon - 1} - (M - m)\right)^2 \leq (y - \tilde{y})^2 \leq \left(\frac{\sum_{\hat{y} \in \hat{\mathcal{Y}}} \hat{y} - |\hat{\mathcal{Y}}|}{e^\epsilon - 1}\right)^2$$

which implies

$$\left(\frac{\sum_{\hat{y} \in \hat{\mathcal{Y}}} \hat{y} - |\hat{\mathcal{Y}}|}{e^\epsilon - 1} - (M - m)\right)^2 \leq 2E_2 = 2\mathbb{E}(y - \tilde{y})^2 \leq \left(\frac{\sum_{\hat{y} \in \hat{\mathcal{Y}}} \hat{y} - |\hat{\mathcal{Y}}|}{e^\epsilon - 1}\right)^2$$

$\square$

For Theorem J.1, the assumptions are reasonable.

- The responses 'ViolentCrimesPerPop' in Communities and Crime dataset, 'SalesAmountInEuro' in the Criteo Sponsored Search Conversion Log Dataset and 'housing price' in California Housing dataset are all nonnegative responses.

- The assumption $\frac{\sum_{\hat{y} \in \hat{\mathcal{Y}}} \hat{y} - |\hat{\mathcal{Y}}|}{e^\epsilon - 1} > (M - m)$ is reasonable. When $\epsilon$ is small, this assumption holds.

For the algorithm by Badanidiyuru et al. (2023), the error bound is very large if $\epsilon$ is small and the average of $\hat{\mathcal{Y}}$ is large. In contrast, our algorithm choose an appropriate interval which dominates most of samples, which avoid a large error bound.

### J.2 TIME AND SPACE COMPLEXITIES

We provide a comparison of time. The time for the methods are almost the same except RRonBins. In the following we report the time (seconds) over 10 trails for all the methods when $\epsilon = 8$ on the the California housing dataset. Similar results for other $\epsilon$ and datasets:

Table 9: Time comparisons.

| Datasets/Method | Laplace | Gaussian | Staircase | RRonBins | RPWithPrior | Unbiased |
|---|---|---|---|---|---|---|
| Housing ($\epsilon = 8$) | 698.82 | 698.7013 | 700.1707 | 768.3160 | 700.7209 | 700.1371 |

- **Discretization in RRonBins**: In our implementation, we discretize the original responses by applying torch.round($y$, decimals $= 2$). We found that using a larger *decimals* resulted in high computational cost, due to the use of dynamic programming. The total runtime for the experiment was approximately 1 hour and 43 minutes when decimals $= 3$.
- **Unbiased**: For the unbiased mechanism proposed by Badanidiyuru et al. (2023), we set the size of both the original and randomized response spaces to 5. The computational cost of the mechanism is highly sensitive to the cardinality of the original and randomized response spaces.

Here we also report the time and space complexities of their differences:

- **RRonBins** (Ghazi et al., 2022): Compute the optimal $\Phi$ (Algorithm 2),
- **RPWithPrior** (ours): Compute Optimal Interval (Algorithm 1),
- **Unbiased** (Badanidiyuru et al., 2023): Compute OptUnbiasedRand$_\epsilon$ (Algorithm 1).

The results are in the following table:

Table 10: Time and space complexities.

| Complexity/Method | RRonBins | RPWithPrior | Unbiased |
|---|---|---|---|
| Time | $O(n_1^3)$ | $O(n_2^2)$ | $O(n_3^{2.5}n_4^{2.5})$ |
| Space | $O(n_1^2)$ | $O(n_2)$ | $O(n_3 n_4)$ |

where $n_1$ is the cardinality of discretized sample space, $n_2$ is the number of intervals and $n_3, n_4$ are the cardinalities of original and randomized spaces.

