# OpenReview forum: "RPWithPrior: Label Differential Privacy in Regression"
_ICLR.cc/2026/Conference — Submitted to ICLR 2026_

### Official Review · Reviewer_dkUH · 2025-10-27

**Soundness:** 3
**Presentation:** 2
**Contribution:** 3
**Rating:** 4
**Confidence:** 3

**Summary:**

This paper addresses the label differential privacy problem in regression tasks and proposes a label encryption mechanism named *RPWithPrior*. The core innovation lies in abandoning the traditional discretization approach and adopting a modeling method based on continuous random variables. Experimental results on real-world datasets demonstrate that the method achieves excellent performance.

**Strengths:**

1. The approach discards the mainstream discretization method and avoids discretization errors by directly modeling continuous random variables.
2. By estimating the optimal interval $[A_1, A_2]$ and constructing a randomized interval, the mechanism can protect the neighborhood relationships of common response values, while allocating probability in the extended interval to meet privacy requirements.
3. Section 4 considers scenarios with an unknown prior distribution, using a histogram perturbed by the Laplace mechanism to approximate $f_Y(\cdot)$, thereby enhancing the method's practicality.
4. Experiments on multiple real-world datasets show that this method consistently achieves top-tier performance.

**Weaknesses:**

1. The current theoretical analysis is insufficient to demonstrate the method's superiority fully.
2. Only experiments on public real-world datasets are presented, simulated data experiments (e.g., generating linear and non-linear datasets) to more intuitively demonstrate the method's effectiveness are not included.

**Questions:**

1. Theoretically, why is it specifically a novel label DP mechanism for regression tasks?  It is recommended to include a more in-depth analysis of theoretical properties.
2. The authors claim the method has lower time complexity. However, the computational complexity analysis in Section G.5 is inadequate. It is recommended to include a table comparing the **theoretical time and space complexity** of various methods.
3. The paper does not detail the tuning strategy for hyperparameters (e.g., $\delta，\sigma$ ). It is recommended to clearly explain the tuning methodology and associated computational cost. As the authors note, a smaller $\sigma$ allows the histogram to more accurately approximate the distribution from the previous stage, but this comes at the cost of increased computational complexity.
4. Regarding the "unbiasedness" mentioned in the conclusion, $\mathbb{E}[\mathcal{M}(y)] = y$. Is this property self-evident? The claim of unbiasedness, $\mathbb{E}[\mathcal{M}(y)] = y$, requires further clarification or proof. Is this property rigorously upheld by the proposed algorithm?
5. Can the optimal interval method (extending from a **one-dimensional interval** to a **multi-dimensional ball**) be more readily generalized to multi-dimensional settings? Could this lead to further theoretical extensions?

minor issue:

6. line 56-59 leaves a large area of blank.

---

> ### Author Response · Authors · 2025-11-15
> **Response to Reviewer dkUH**
>
> Dear Reviewer dkUH,
>
> Thank you for your thoughtful review and constructive comments on our paper.
>
> **Q1**: why is it specifically a novel label DP mechanism for regression tasks? It is recommended to include a more in-depth analysis of theoretical properties.
>
> **A**: We compare those existing methods and ours as follows:
>
> In machine learning theory, **regression** and **classification** are distinct tasks, defined by their output spaces:
>
> **Regression** addresses estimation problems where the output space $Y$ is **continuous** and ordered, with the goal of predicting a real-valued quantity.
>
> **Classification** deals with problems where the output space is a **finite** set of **discrete** categories.
>
> The methods of Ghazi et al. (2022) and Badanidiyuru et al. (2023) fundamentally **convert a regression problem into a classification problem** within the framework of Label Differential Privacy. For RRonbins, this conversion is achieved through a multi-step process:
>
> **Rounding**: The original continuous responses (e.g., the Criteo Search dataset) are first rounded to integers.
>
> **Discretization (Binning)**: A dynamic programming algorithm is used to partition the rounded values into a finite set of bins, effectively creating a set of discrete classes.
>
> **Randomization**: A standard Randomized Response (RR) mechanism is applied over this finite set of bins.
>
> In contrast, **our method directly** addresses the regression problem under Label DP.
>
>
> **Q2**: The authors claim the method has lower time complexity. However, the computational complexity analysis in Section G.5 is inadequate. It is recommended to include a table comparing the theoretical time and space complexity of various methods.
>
> **A**: In the following we report the time (seconds) over 10 trails for all the methods when $\epsilon=8$. We find they are almost the same except RRonBins. Similar results for other $\epsilon$ and datasets:
> | Datasets/Method|Laplace|Gaussian|Staircase|RRonBins| Our| Unbiased|
> |---------|-------|-------|-------|-------|-------|-------|
> |Housing| 698.82|698.7013|700.1707|768.3160|700.7209|700.1371|
>
> **Discretization in RRonBins**:
> In our implementation, we discretize the original responses by applying torch.round(y, decimals=2). We found that using a larger *decimals* resulted in high computational cost, due to the use of dynamic programming. The total runtime for the experiment was approximately 1 hour and 43 minutes when decimals=3.
>
> **Unbiased**:
> For the unbiased mechanism proposed by Badanidiyuru et al. (2023), we set the size of both the original and randomized response spaces to 5. The computational cost of the mechanism is highly sensitive to the cardinality of the original and randomized response
> spaces.
>
> Here we also report the time and space complexities of their differences
>
> - Compute the optimal $\Phi$ (Algorithm 2),
> - Compute Optimal Interval (Algorithm 1),
> - Compute OptUnbiasedRand$_\epsilon$ (Algorithm 1).
>
> The results are in the following table:
>
> | Complexity/Method | RRonbins    | Our     | Unbiased     |
> |---------------|---------|---------|---------|
> | Time         | $O(n^3_1)$  |  $O(n^2_2)$ | $O(n^{2.5}_3 n^{2.5}_4)$  |
> | Space         | $O(n^2_1)$  | $O(n_2)$  | $O(n_3 n_4)$  |
>
> where $n_1$ is the unique number of discretized samples, $n_2$ is the number of intervals and $n_3,n_4$ are the numbers of the original and randomized spaces.
>
> **Q3.1**: The paper does not detail the tuning strategy for hyperparameters (e.g., $\delta,\sigma$). It is recommended to clearly explain the tuning methodology and associated computational cost. As the authors note, a smaller
>  allows the histogram to more accurately approximate the distribution from the previous stage, but this comes at the cost of increased computational complexity.
>
> **A**: The performance of our method is largely insensitive to the value of $\sigma$. In empirical tests, varying $\sigma$ by a factor of $\alpha$ around the fixed values $\lbrace 0.5, 1, 2\rbrace$ resulted in negligible change to the outcome. This robustness is expected, as Algorithm 4 is used solely to estimate the prior distribution for determining the optimal interval.
>
>  The value of $\delta$ for different datasets (also see Appendix G.4):
>
> | dataset/$\epsilon$ | 0.05    | 0.1     | 0.3     | 0.5     | 0.8     | 1       | 1.5     | 2       | 3       | 4       | 6       | 8       |$\infty$       |
> |---------------|---------|---------|---------|---------|---------|---------|---------|---------|---------|---------|---------|---------|---------|
> | Crime         | 0.4  | 0.4  | 0.4  | 0.4  | 0.4  | 0.4  | 0.4  | 0.4  | 0.4  | 0.4  | 0.4  | 0.4  |0.4  |
> | Criteo         | 70  | 70  | 70  | 70  | 70  | 70  | 70  | 70  | 70  | 70  | 70  | 70  |70  |
> | Housing        | 0.7  | 0.5  | 1.2  | 1.0  | 2.2  | 1.5  | 1.5  | 1.5  | 1.4  | 1.2  | 0.7  | 0.1  |0.1  |
>
> We choose parameters manually, similar as grid search. From the above table, we find it is not sensitive of the parameter $\delta$.

---

> > ### Author Response · Authors · 2025-11-15
> > **Response to Reviewer dkUH**
> >
> > **Q3.2**: As the authors note, a smaller allows the histogram to more accurately approximate the distribution from the previous stage, but this comes at the cost of increased computational complexity.
> >
> > **A**: The following table is about the privacy budget allocated for prior estimation:
> >
> > | dataset/$\epsilon$ | 0.05    | 0.1     | 0.3     | 0.5     | 0.8     | 1       | 1.5     | 2       | 3       | 4       | 6       | 8       |$\infty$       |
> > |---------------|---------|---------|---------|---------|---------|---------|---------|---------|---------|---------|---------|---------|---------|
> > | Crime         | $\epsilon/2$  | $\epsilon/2$  | $\epsilon/2$  | $\epsilon/2$  | $\epsilon/2$  | $\epsilon/2$  | $\epsilon/2.5$  | $\epsilon/2.5$  | $\epsilon/3$  | $\epsilon/3.5$  | $\epsilon/6$  | $\epsilon/7$  |$\infty$  |
> > | Criteo         | 0.017  | 0.017  | 0.017  | 0.017  | 0.017  | 0.017  | 0.01  | 0.01  | 0.01  | 0.005  | 0.003  | 0.002  |$\infty$  |
> > | Housing        |  0.017  | 0.017  | 0.017  | 0.017  | 0.01  | 0.008  | 0.008  | 0.008  | 0.007  | 0.007  | 0.007  | 0.007  |$\infty$  |
> >
> > which decreases as the total privacy budget increases. This indicates that a highly accurate prior estimation, derived from the noisy data itself, is not crucial for determining the optimal interval under stronger privacy guarantees.
> >
> > Furthermore, we observe a clear empirical relationship between the privacy budget and the optimal interval: a smaller privacy budget $\epsilon$ leads to a smaller optimal interval, and vice versa.
> >
> > **Q4.1**: Regarding the "unbiasedness" mentioned in the conclusion, $\mathbb{E}(\mathcal{M}(y))=y$. Is this property self-evident? The claim of unbiasedness, $\mathbb{E}(\mathcal{M}(y))=y$, requires further clarification or proof.
> >
> > **A**: In classical statistics, an estimator is unbiased if its expected value equals the true parameter value, i.e., $\mathbb{E}[\hat{\theta}] = \theta$. A natural extension for a randomized mechanism $\mathcal{M}$ would be $\mathbb{E}[\mathcal{M}(y)] = y$ for all $y \in \mathcal{Y}$. However, a rigorous and universally adopted definition of unbiasedness within the framework of differential privacy is currently lacking.
> >
> > **Q4.2**: Is this property rigorously upheld by the proposed algorithm?
> >
> > **A**: Our proposed algorithm solves the optimization problem (3.3). To ensure the mechanism is unbiased, the constraint could be incorporated directly into the problem's formulation.
> >
> > **Q5**: Can the optimal interval method (extending from a one-dimensional interval to a multi-dimensional ball) be more readily generalized to multi-dimensional settings? Could this lead to further theoretical extensions?
> >
> > **A**: Yes, the framework can be extended to the multivariate case. The most straightforward approach is to assume the perturbation mechanisms across dimensions are independent. Formally, we assume the conditional density factorizes as:
> > $$f(\tilde{y}|y)=\Pi^k_{i=1}f_i(\tilde{y}_i|y_i).$$
> > Under this assumption, we can apply a one-dimensional perturbation mechanism to each dimension $y_i$ separately and independently. This principle of independent noise across dimensions is, in fact, the standard approach used by additive noise mechanisms like the multivariate Laplace or Gaussian mechanism.
> >
> > **Q6**: minor issue: line 56-59 leaves a large area of blank.
> >
> > **A**: Thank you for the comment. We updated.

---

> ### Author Response · Authors · 2025-11-17
> **Reply to Reviewer dkUH**
>
> Dear Reviewer dkUH,
>
> Thank you for your time and efforts to review our paper. The following are the reply to the weakness you mentioned.
>
> **W1**: The current theoretical analysis is insufficient to demonstrate the method's superiority fully.
>
> **A**: Thank you for the comment.
>
> The methods of Ghazi et al. (2022) and Badanidiyuru et al. (2023) fundamentally **convert a regression problem into a classification problem** within the framework of Label Differential Privacy.
>
> In contrast, **our method directly** addresses the regression problem under Label DP.
>
> In machine learning theory, **regression** and **classification** are distinct tasks, defined by their output spaces:
>
> **Regression** addresses estimation problems where the output space $Y$ is **continuous** and ordered, with the goal of predicting a real-valued quantity.
>
> **Classification** deals with problems where the output space is a **finite** set of **discrete** categories.
>
> For Ghazi et al. (2022), this conversion is achieved through a multi-step process:
>
> **Rounding**: The original continuous responses (e.g., the Criteo Search dataset) are first rounded to integers.
>
> **Discretization (Binning)**: A dynamic programming algorithm is used to partition the rounded values into a finite set of bins, effectively creating a set of discrete classes.
>
> **Randomization**: A standard Randomized Response (RR) mechanism is applied over this finite set of bins.
>
> For Badanidiyuru et al. (2023), the conversion is as follows:
>
> **unbiased Randomized rounding to finite points**:
>
> **discrete mapping**: construct a discrete randomized response space as follows:
> $$L=((e^\epsilon+|\mathcal{Y}|-1)\min(\mathcal{Y})-\sum_{y\in\mathcal{Y}}y)/(e^\epsilon-1)$$
> $$U=((e^\epsilon+|\mathcal{Y}|-1)\max(\mathcal{Y})-\sum_{y\in\mathcal{Y}}y)/(e^\epsilon-1)$$
> $$\Delta=(U-L)/(n-1)$$
> where $\mathcal{Y}$ is the discretized space of the original responses. The randomized response space is
> $$\hat{\mathcal{Y}}=(L,L+\Delta,\cdots,U-\Delta,U)$$
> which contains $n$ discrete points.
>
> **Randomization**: compute the transition matrix from the original responses to the randomized responses and randomization.
>
> In contrast, **our method directly** addresses the regression problem under Label DP.
>
> **No Discretization**: Our methodology is fundamentally continuous. The original responses are treated as continuous random variables and are never discretized at any stage of the process.
>
> **Theoretical Support**: The primary contribution of our theoretical results is to establish the foundation for this continuous approach. They rigorously support an algorithm designed to map a continuous random variable to another continuous random variable while satisfying the required privacy constraints.

---

> ### Author Response · Authors · 2025-11-17
> **Reply to Reviewer dkUH**
>
> **W2**: simulated data experiments (linear and non-linear datasets).
>
> **A**: We do numerical experiments on simulated dataset (both linear and nonlinear) as follows: let $X\in\mathbb{R}^{n\times d}$, $y\in\mathbb{R}^{n}$, where
> $n$ is the sample size and $d$ is the dimension. In the both cases, we generate the feature matrix $X$ randomly obeying Standard Normal distribution.
>
> **Linear**: we also generate the vector $A\in\mathbb{R}^d$ randomly obeying Standard Normal distribution and set $y=\sum^d_{i=1}A[i]X[:,i]$, where $X[:,i]$ is the $i$-th column of $X$. The numerical results are as follows:
>
> | Method/$\epsilon$ | 0.05|0.1|0.3| 0.5| 0.8|1| 1.5 | 2 | 3 |4| 6|8|$\infty$|
> |-------------------------|-----------------|-----------------|-----------------|-----------------|-----------------|-----------------|-----------------|-----------------|-----------------|-----------------|-----------------|-----------------|-----------------|
> | Laplace| 6.7619$\pm$3.0561  | 11.6518$\pm$6.5433  | 10.5618$\pm$4.6079  | 8.3280$\pm$5.66974  | 7.2296$\pm$3.2733  | 6.4560$\pm$2.9871  | 5.6707$\pm$3.4424  | 7.9935$\pm$4.1851  | 4.0967$\pm$1.2403  | 4.6560$\pm$2.1379  | 2.7771$\pm$1.2930  | 3.3923$\pm$2.1478  |0.0120$\pm$0.0050  |
> | Gaussian| 10.8907$\pm$3.8553 | 11.2220$\pm$3.1782  | 10.7778$\pm$3.5415  | 9.1884$\pm$5.1005  | 7.4162$\pm$3.9262  | 10.2868$\pm$3.1686  | 7.8765$\pm$3.3130  | 9.5327$\pm$4.7792  | 7.0141$\pm$5.1296  | 4.4751$\pm$2.5818  | 6.3369$\pm$4.1048  | 5.2665$\pm$3.3694  | 0.0120$\pm$0.0050 |
> | Staircase|  12.1042$\pm$3.3601  | 9.9774$\pm$5.1210  | 7.7710$\pm$3.4123  | 9.1060$\pm$3.0008  | 4.6615$\pm$2.4509  | 6.1859$\pm$4.6726  | 3.9385$\pm$2.9584  | 6.2754$\pm$3.5710  | 5.9760$\pm$2.3838  | 5.0915$\pm$3.5990  | 4.5594$\pm$2.9253  | 3.9599$\pm$1.8278  |0.0120$\pm$0.0050  |
> | RRonbins| 10.1382$\pm$4.3226  | 10.2426$\pm$3.7583  | 8.7568$\pm$3.2488  | 10.5588$\pm$3.4701  | 7.4461$\pm$3.7534  | 9.3996$\pm$1.7144  | 5.7334$\pm$2.2097  | 5.7551$\pm$2.4432  | 5.1670$\pm$1.5725  | 5.5471$\pm$2.3634  | 5.2050$\pm$3.7011  | 5.1476$\pm$3.1585  |0.0173$\pm$0.0053  |
> | Ours|  9.2342$\pm$4.7130  | 5.7169$\pm$1.5244  | 10.3128$\pm$5.899  | 6.7450$\pm$3.6947  | 8.7830$\pm$3.8595  | 7.2708$\pm$2.5645  | 5.2094$\pm$2.2437  | 4.8456$\pm$3.1561  | 5.1234$\pm$1.3356  | 3.7567$\pm$3.0159  | 3.6992$\pm$2.9984  |4.1412$\pm$1.8096  |0.0125$\pm$0.0069  |
> | Unbiased|11.8071$\pm$5.0206|10.4242$\pm$4.9875|9.9343$\pm$4.0548|10.5926$\pm$3.7888| 9.3589$\pm$4.1997  | 7.2313$\pm$3.9793  | 7.3531$\pm$3.1218|5.1702$\pm$2.2089|8.6822$\pm$5.4475| 4.7316$\pm$2.0718|4.1146$\pm$2.7303|4.6723$\pm$2.8677|0.0219$\pm$0.0086 |
>
> For linear dataset, all the methods can get better results.
>
> **Nonlinear**: $y=\sum^d_{i=1}\exp(X[:,i])$, where $X[:,i]$ is the $i$-th column of $X$. The numerical results are as follows:
>
> | Method/$\epsilon$ | 0.05|0.1| 0.3 | 0.5| 0.8|1|1.5| 2 |3 |4 |6|8|$\infty$|
> |---------------------|---------------|---------------|---------------|---------------|---------------|---------------|---------------|---------------|---------------|---------------|---------------|---------------|---------------|
> | Laplace|12187.3481$\pm$13439.9494|3565.2386$\pm$2162.5579|392.2554$\pm$182.3960| 162.0974$\pm$76.9350|79.2950$\pm$26.7152|**61.0212$\pm$42.8874**|**40.5246$\pm$24.7389**| **25.0478$\pm$6.4657**|**15.2806$\pm$3.3548**|**13.1973$\pm$1.4805**|**11.9794$\pm$2.1062**| **9.4974$\pm$3.0906**|**6.4681$\pm$1.9020**|
> | Gaussian| 39383.0625$\pm$22130.0212|8309.4118$\pm$4890.6477|3090.6682$\pm$1624.5669| 1684.8579$\pm$1107.2722|426.5728$\pm$246.2399| 289.4453$\pm$116.5660|**200.5178$\pm$195.2500**| 83.4924$\pm$30.2726|56.2989$\pm$38.6322| 34.6206$\pm$10.0915  | 20.6233$\pm$5.2040  | 17.3384$\pm$3.6793  |**6.4681$\pm$1.9020**|
> | Staircase|  7379.2940$\pm$3660.3196|1871.4528$\pm$817.2671| 513.4610$\pm$383.4003| **132.1034$\pm$76.9102**| **53.2645$\pm$23.1531**| 48.6851$\pm$28.3232| **37.3397$\pm$28.0381**| **18.0702$\pm$3.9350**| **16.1297$\pm$2.7485**| 17.7200$\pm$2.1758|19.2643$\pm$4.1972|19.6195$\pm$3.9102  |**6.4681$\pm$1.9020**|
> | RRonbins| **312.2784$\pm$115.1406**|451.3280$\pm$184.7202|348.5720$\pm$131.7954| 340.9608$\pm$138.6306|344.2565$\pm$189.8461|236.5660$\pm$76.9058|352.5059$\pm$168.2677| 221.9705$\pm$75.8004|91.2669$\pm$24.0746| 60.4433$\pm$20.9969| 25.6626$\pm$10.1864| **11.8751$\pm$2.8385**|**5.9318$\pm$1.5030**|
> | Ours|  **155.4341$\pm$76.1839**| **53.9678$\pm$22.0496**| **42.0532$\pm$3.4690**| **40.4100$\pm$4.0826**  | **37.4296$\pm$3.8043**| **17.0438$\pm$1.7695**| **26.2684$\pm$4.4789**| **21.3296$\pm$2.0643**| **16.5606$\pm$1.4494**| **15.4449$\pm$2.0659**| **12.587$\pm$5.8092**| **12.6356$\pm$2.6500**|  **6.6823$\pm$1.6788**|
> | Unbiased|1252.6487$\pm$537.9432|796.6692$\pm$384.3283|2456.8288$\pm$3920.4370| 563.4305$\pm$201.8678| 731.2202$\pm$430.8711  | 665.3807$\pm$247.5771  | 414.7472$\pm$169.2534|396.3458$\pm$246.6874| 156.4980$\pm$71.7431 |105.0789$\pm$38.5969|34.0717$\pm$22.5132| **15.9530$\pm$3.6365**  |**7.2015$\pm$1.3027**|

---

> ### Comment · Reviewer_dkUH · 2025-11-26
> **Response to the author**
>
> Thank you for the authors' response, which has addressed my concerns to some extent. The innovation of continuous relaxation lacks theoretical guarantees and relies primarily on empirical results. As another anonymous reviewer pointed out, the comparative advantages over the work of Badanidiyuru et al. (2023) are not sufficiently pronounced. The paper may need to restructure its descriptions to better highlight its own innovations and distinctions from related works.

---

> ### Author Response · Authors · 2025-11-27
> **Reply to Reviewer dkUH**
>
> Dear Reviewer dkUH,
>
> Please accept our sincere gratitude for your thoughtful review of our paper, we will reconstruct our revision. In the following we provide the theoretical and numerical results.
>
> **Theoretical Results**:
>
> **Theorem**: We consider a nonnegative response space and assume $\frac{\sum_{\hat y\in\hat{\mathcal{Y}}}\hat y-|\hat{\mathcal{Y}}|}{e^\epsilon-1}>(M-m)$, where $\hat{\mathcal{Y}}$ is the unbiased randomized space, $|\hat{\mathcal{Y}}|$ is the cardinality of $\hat{\mathcal{Y}}$ and $m,M$ are the minimum and maximum values of $\hat{\mathcal{Y}}$.  Let $\Omega=[0,1]^d$ and $\ell(\cdot,\cdot)$ be $L_p$ norm, $x,y,\tilde y$ be a predictor, a true response and a randomized response, respectively. Let $\tilde g(x)$ be a predicted regression function by $(x,\tilde y)$ and $g(x)\in L_p(\Omega)$ be a true regression function such that $g(x)=\tilde y$. Then the error can be bounded as
> $$\mathbb E_{x,y,\tilde y}[\ell(\tilde g(x),y)]\le 2C||f||L^{-2s/d}+\left(\frac{\sum_{\hat y\in\hat{\mathcal{Y}}}\hat{y}-|\hat{\mathcal{Y}}|}{e^\epsilon-1}\right)^2,$$
> where $||f||=||f||_{W^s(L_q(\Omega))}$
>
> or
>
> $||f||=||f||_{B^s_r(L_q(\Omega))}.$
>
> **Proof**:
> $$\mathbb E_{x,y,\tilde y}[\ell(\tilde g(x),y)]\le 2\mathbb E_{x,y,\tilde y}[\ell(\tilde g(x),\tilde y)]+2\mathbb{E}_{x,y,\tilde y}[\ell(\tilde y,y)]:=2E_1+2E_2.$$
>
> In deep learning, $E_1$ (model risk) decays as network complexity increases. From [1], we have
> $$E_1\le C||f||L^{-2s/d}$$
>  for a constant $C:=C(s,q,p,d)<+\infty$.
>
> Next, we estimate the upper and lower bounds of $E_2$. Since $\frac{\sum_{\hat y\in\hat{\mathcal{Y}}}\hat y-|\hat{\mathcal{Y}}|}{e^\epsilon-1}>(M-m)$, we have $M-\frac{\sum_{\hat y\in\hat{\mathcal{Y}}}\hat y-|\hat{\mathcal{Y}}|}{e^\epsilon-1}<m$. Note that $\mathcal{Y}$ and $\hat{\mathcal{Y}}$ share the same minimum and maximum value. That is, the minimum of the original response space $\hat{\mathcal{Y}}$ is larger than the maximum of the randomized response space $\tilde{\mathcal{Y}}$. Therefore,
>
> $$\left(\frac{\sum_{\hat y\in\hat{\mathcal{Y}}}\hat y-|\hat{\mathcal{Y}}|}{e^\epsilon-1}-(M-m)\right)^2\le(y-\tilde y)^2\le\left(\frac{\sum_{\hat y\in\hat{\mathcal{Y}}}\hat y-|\hat{\mathcal{Y}}|}{e^\epsilon-1}\right)^2$$
> which implies
> $$\left(\frac{\sum_{\hat y\in\hat{\mathcal{Y}}}\hat y-|\hat{\mathcal{Y}}|}{e^\epsilon-1}-(M-m)\right)^2\le2E_2=2\mathbb{E}(y-\tilde y)^2\le\left(\frac{\sum_{\hat y\in\hat{\mathcal{Y}}}\hat y-|\hat{\mathcal{Y}}|}{e^\epsilon-1}\right)^2$$
>
> **Note**:
> - The responses `ViolentCrimesPerPop` in Communities and Crime dataset, `SalesAmountInEuro` in the Criteo Sponsored Search Conversion Log Dataset and `housing price` in California Housing dataset are all nonnegative responses.
>
> - The assumption $\frac{\sum_{\hat y\in\hat{\mathcal{Y}}}\hat y-|\hat{\mathcal{Y}}|}{e^\epsilon-1}>(M-m)$ is reasonable. When $\epsilon$ is small, this assumption holds. Hence, the error bound is very large if the average of $\hat{\mathcal{Y}}$ is large.
>
> - In contrast, our algorithm choose an appropriate interval which dominates most of samples, which avoid a large error bound.
>
> [1] J. W. Siegel. Optimal approximation rates for deep relu neural networks on sobolev and besov spaces. Journal
> of Machine Learning Research, 24(357):1–52, 2023.
>
> **Numerical results** of the Criteo Sponsored Search Conversion Log Dataset for the algorithm by Badanidiyuru et al. (2023):
>
>
> | epsilon/dataset | Criteo     |
> |---------------|---------|
> | 0.05         |  10572.5890$\pm$1134.12  |
> | 0.1          | 7884.54$\pm$664.82  |
> | 0.3          | 5807.11$\pm$393.60  |
> | 0.5           | 4826.07$\pm$250.91  |
> | 0.8           | 4769.94$\pm$210.42  |
> | 1           | 4630.83$\pm$171.56  |
> | 1.5           | 4472.40$\pm$92.37  |
> | 2           | 4459.39$\pm$78.33  |
> | 3             | 4330.81$\pm$58.60  |
> | 4           | 4244.34$\pm$76.74  |
> | 6           | 3241.17$\pm$44.89  |
> | 8           | 2911.97$\pm$58.30  |
> | $\infty$     | 2798.53$\pm$29.11 |
>
> - The numerical results are sensitive to parameters. The results will get much worse even we change the batchsize to 8192.

---

### Official Review · Reviewer_nRv3 · 2025-10-29

**Soundness:** 1
**Presentation:** 1
**Contribution:** 2
**Rating:** 2
**Confidence:** 3

**Summary:**

This paper introduces an extension of RPWithPrior to continuous responses, achieving label differential privacy (label DP) in regression tasks. Unlike existing methods that discretize the continuous output/label space into bins, this work models both the original and randomized labels as continuous random variables. The core idea is to define an optimal interval [A1, A2] based on a prior distribution of the labels; the mechanism then decides whether the response should lie within this interval or not. The authors provide theoretical proofs for the $\epsilon$-label DP guarantee and demonstrate through experiments on three datasets that their method outperforms baseline mechanisms (Gaussian, Laplace, Staircase, and RRonBins) in terms of test Mean Squared Error (MSE).

**Strengths:**

1. This paper proposes a new label-DP mechanism for continuous random variables.

2. The theoretical results are straightforward and seem correct, even though I have not checked the proofs.

3. The experiments are clean, and the proposed mechanism improves empirical performance according to the experimental results on the Criteo Sponsored Search Conversion Log dataset.

**Weaknesses:**

1. The contributions of this paper are significantly over-claimed. In fact, the main contribution is proposing a new discretization of a continuous random variable and then applying Randomized Response (RR) to the discretized response. However, the authors claim that "this work presents the first introduction of continuous random variables," despite similar discretization methods already existing in prior research, such as Badanidiyuru et al., 2023. The abstract and introduction repeat the same high-level contributions multiple times, which is redundant and frustrating.

2. The introduction is poorly written and contains limited information. Furthermore, some mathematical definitions should be moved to a preliminary section.

3. The main contribution is poorly written, making it difficult to understand. Many mathematical notations are not explained.

a) Please explain the motivation for formalizing the problem as (3.3).

b) The objective function $F(A_1,A_2)$ is not introduced when it first appears in Section 3.2. The authors should also explain why it takes the form given in Lemma 3.5.

c) The term "optimal interval" is not clearly defined.

4.The conclusion is not self-contained. The mathematical writing is dense and often inconsistent, creating unnecessary barriers. For example, the authors claim to focus on continuous responses, yet in Lemma 3.5, they assume the pdf of $y$ is a step function, implying a discrete distribution. This inconsistency may undermine the paper's main contribution.

5.  The choice of the $\delta$ parameter for the intervals seems very important, but the authors do not provide details on how to select it. Moreover, in the DP literature, $\delta$ typically refers to the parameter in $(\epsilon,\delta)$-DP, so it would be better to use a different Greek letter.

6. The method for estimating an unknown prior relies on a simple histogram of the Laplace-noised data. This approach can be inaccurate is sensitive to the histogram's bin width ($\sigma$). While Remark 4.1 acknowledges this trade-off, the paper does not explore more sophisticated density estimation techniques that could improve performance.

7. The theoretical analysis relies on one-dimensional calculus, and as noted, discretizing a continuous response is not novel. Therefore, the theoretical contribution is insufficient. My publication recommendation mainly relies on whether the empirical contribution is strong enough.

8.  The empirical contribution is too limited, which is one of the main reasons for my rating.

a) First, the experiments provide limited comparisons to state-of-the-art methods. The authors cite Badanidiyuru et al., 2023 but do not include it in their experimental comparisons.

b) Second, the experiments are conducted on only one simple real-world dataset. Compared to existing works like Badanidiyuru et al., 2023, the experimental investigation is inadequate.

The authors should compare their method to Badanidiyuru et al., 2023 on more datasets.

**Questions:**

See the weaknesses part

---

> ### Author Response · Authors · 2025-11-15
> **Response to Reviewer nRv3**
>
> Dear Reviewer nRv3,
>
> Thank you for your thoughtful review and constructive comments on our paper.
>
> **W1**: The contributions of this paper are significantly over-claimed. In fact, the main contribution is proposing a new discretization of a continuous random variable and then applying Randomized Response (RR) to the discretized response. However, the authors claim that "this work presents the first introduction of continuous random variables," despite similar discretization methods already existing in prior research, such as Badanidiyuru et al., 2023. The abstract and introduction repeat the same high-level contributions multiple times, which is redundant and frustrating.
>
> **A**: We apologize for any confusion. In our method, both the original and randomized responses are treated as continuous variables. The randomized responses belong to a continuous interval $[A_1-\delta,A_2+\delta]$, not a finite set of discrete points.
>
> This contrasts with the approach of Badanidiyuru et al. (2023), who first discretize the original responses and then construct a discrete randomized response space as follows:
> $$L=((e^\epsilon+|\mathcal{Y}|-1)\min(\mathcal{Y})-\sum_{y\in\mathcal{Y}}y)/(e^\epsilon-1)$$
> $$U=((e^\epsilon+|\mathcal{Y}|-1)\max(\mathcal{Y})-\sum_{y\in\mathcal{Y}}y)/(e^\epsilon-1)$$
> $$\Delta=(U-L)/(n-1)$$
> where $\mathcal{Y}$ is the discretized space of the original responses. The randomized response space is
> $$\hat{\mathcal{Y}}=(L,L+\Delta,\cdots,U-\Delta,U)$$
> which contains $n$ discrete points.
>
> We update this part in the introduction section as follows: **this work introduces continuous random
> variables for non-additive mechanisms in label DP.**
>
> **W2**: The introduction is poorly written and contains limited information. Furthermore, some mathematical definitions should be moved to a preliminary section.
>
> **A**: In the introduction section, we mainly review the problems of DP, Label DP, their applications, some limitations of some existing algorithms and introduce our proposed algorithm and contributions.
>
> We move the mathematical definitions to a preliminary section, thank you for the comment.
>
> **W3.1**: The main contribution is poorly written, making it difficult to understand. Many mathematical notations are not explained. a) Please explain the motivation for formalizing the problem as (3.3).
>
> **A**: Our method is motivated by the empirical distribution of the responses. Since most responses are concentrated within a finite interval, with only a few extreme outliers, we seek an optimal interval $\mathcal{I}$. This interval maximizes the probability that a response $y$ is mapped to a noisy value $\tilde{y}$ within its neighborhood $\mathcal{N}_y$, for all $y \in \mathcal{I}$, while still satisfying $\epsilon$-label differential privacy.
>
> **W3.2**: b) The objective function $F(A_1,A_2)$ is not introduced when it first appears in Section 3.2. The authors should also explain why it takes the form given in Lemma 3.5.
>
> **A**: $F(A_1,A_2)$ is the maximal value of the objective function in the optimization problem (3.3), which is guaranteed in Lemma 3.3.
>
> **W3.3**: c) The term "optimal interval" is not clearly defined.
>
> **A**: For any given interval $[A_1, A_2]$, $F(A_1, A_2)$ denote the maximum achievable value of the objective function in (3.3). Our goal is to find the optimal interval by maximizing $F(A_1, A_2)$ over all possible $A_1$ and $A_2$, where the optimal solution corresponds to the endpoints of the optimal interval.
>
> **W4**: The conclusion is not self-contained. The mathematical writing is dense and often inconsistent, creating unnecessary barriers. For example, the authors claim to focus on continuous responses, yet in Lemma 3.5, they assume the pdf of $y$ is a step function, implying a discrete distribution. This inconsistency may undermine the paper's main contribution.
>
> **A**: We regret any misunderstanding. A continuous random variable takes values in a continuous space (e.g., an interval $[A_1,A_2]$), not a finite set. Its probability density function (pdf) is any integrable, non-negative function that integrates to 1 over this space; the pdf itself may be continuous or not.

---

> > ### Author Response · Authors · 2025-11-15
> > **Response to Reviewer nRv3**
> >
> > **W5**: The choice of the $\delta$ parameter for the intervals seems very important, but the authors do not provide details on how to select it. Moreover, in the DP literature, $\delta$ typically refers to the parameter in $(\epsilon,\delta)$-DP, so it would be better to use a different Greek letter.
> >
> > **A**:  The value of $\delta$ for different datasets (also see Appendix G.4):
> >
> > | dataset/$\epsilon$ | 0.05    | 0.1     | 0.3     | 0.5     | 0.8     | 1       | 1.5     | 2       | 3       | 4       | 6       | 8       |$\infty$       |
> > |---------------|---------|---------|---------|---------|---------|---------|---------|---------|---------|---------|---------|---------|---------|
> > | Crime         | 0.4  | 0.4  | 0.4  | 0.4  | 0.4  | 0.4  | 0.4  | 0.4  | 0.4  | 0.4  | 0.4  | 0.4  |0.4  |
> > | Criteo         | 70  | 70  | 70  | 70  | 70  | 70  | 70  | 70  | 70  | 70  | 70  | 70  |70  |
> > | Housing        | 0.7  | 0.5  | 1.2  | 1.0  | 2.2  | 1.5  | 1.5  | 1.5  | 1.4  | 1.2  | 0.7  | 0.1  |0.1  |
> >
> > We choose parameters manually, similar as grid search.
> >
> > From the above table, we find it is not sensitive of the parameter $\delta$.
> >
> > We thank the reviewer for pointing this out. We will revise the manuscript to swap the notation, using $\delta$ to denote the differential privacy parameter and $\zeta$ for the superparameter, to align with common convention and avoid confusion.
> >
> > **W6**: The method for estimating an unknown prior relies on a simple histogram of the Laplace-noised data. This approach can be inaccurate is sensitive to the histogram's bin width ($\delta$). While Remark 4.1 acknowledges this trade-off, the paper does not explore more sophisticated density estimation techniques that could improve performance.
> >
> > **A**: The privacy budget allocated for prior estimation decreases as the total privacy budget increases (see the following table, also reported in Appendix G.4).
> >
> > | dataset/$\epsilon$ | 0.05    | 0.1     | 0.3     | 0.5     | 0.8     | 1       | 1.5     | 2       | 3       | 4       | 6       | 8       |$\infty$       |
> > |---------------|---------|---------|---------|---------|---------|---------|---------|---------|---------|---------|---------|---------|---------|
> > | Crime         | $\epsilon/2$  | $\epsilon/2$  | $\epsilon/2$  | $\epsilon/2$  | $\epsilon/2$  | $\epsilon/2$  | $\epsilon/2.5$  | $\epsilon/2.5$  | $\epsilon/3$  | $\epsilon/3.5$  | $\epsilon/6$  | $\epsilon/7$  |$\infty$  |
> > | Criteo         | 0.017  | 0.017  | 0.017  | 0.017  | 0.017  | 0.017  | 0.01  | 0.01  | 0.01  | 0.005  | 0.003  | 0.002  |$\infty$  |
> > | Housing        |  0.017  | 0.017  | 0.017  | 0.017  | 0.01  | 0.008  | 0.008  | 0.008  | 0.007  | 0.007  | 0.007  | 0.007  |$\infty$  |
> >
> > This indicates that a highly accurate prior estimation, derived from the noisy data itself, is not crucial for determining the optimal interval under stronger privacy guarantees.
> >
> > Furthermore, we observe a clear empirical relationship between the privacy budget and the optimal interval: a smaller privacy budget $\epsilon$ leads to a smaller optimal interval, and vice versa.
> >
> > **W7**: The theoretical analysis relies on one-dimensional calculus, and as noted, discretizing a continuous response is not novel. Therefore, the theoretical contribution is insufficient. My publication recommendation mainly relies on whether the empirical contribution is strong enough.
> >
> > **A**: We would like to clarify two key aspects to prevent any misunderstanding:
> >
> > **No Discretization**: Our methodology is fundamentally continuous. The original responses are treated as continuous random variables and are never discretized at any stage of the process.
> >
> > **Theoretical Support**: The primary contribution of our theoretical results is to establish the foundation for this continuous approach. They rigorously support an algorithm designed to map a continuous random variable to another continuous random variable while satisfying the required privacy constraints.

---

> ### Author Response · Authors · 2025-11-15
> **Response to Reviewer nRv3**
>
> **W8.1**: The empirical contribution is too limited, which is one of the main reasons for my rating. First, the experiments provide limited comparisons to state-of-the-art methods. The authors cite Badanidiyuru et al., 2023 but do not include it in their experimental comparisons.
>
> **A**: We provide the comparison results as follows:
>
> | epsilon/dataset | Crime    | Criteo     | Housing     |
> |---------------|---------|---------|---------|
> | 0.05         | 0.1293$\pm$0.0096  | 21177.52$\pm$250.42  | 1.6307$\pm$0.1155  |
> | 0.1          | 0.1194$\pm$0.0056  | 20784.56$\pm$142.07  | 1.6554$\pm$0.0542  |
> | 0.3          | 0.1110$\pm$0.0094  | 18937.41$\pm$140.36  | 1.5656$\pm$0.0625  |
> | 0.5           | 0.1037$\pm$0.0089  | 17432.69$\pm$197.66  | 1.4666$\pm$0.0644  |
> | 0.8           | 0.0892$\pm$0.0064  | 15221.72$\pm$177.23  | 1.42419$\pm$0.0700  |
> | 1           | 0.0826$\pm$0.0039  | 13910.56$\pm$204.06  | 1.3641$\pm$0.0617  |
> | 1.5           | 0.0668$\pm$0.0041  | 10876$\pm$222.24  | 1.1896$\pm$0.0447  |
> | 2           | 0.0521$\pm$0.0032  | 8622.52$\pm$120.50  | 1.0882$\pm$0.0657  |
> | 3             | 0.0355$\pm$0.0021  | 5584.71$\pm$61.14  | 0.9587$\pm$0.0766  |
> | 4           | 0.0259$\pm$0.0024  | 4110.17$\pm$53.86  | 0.7659$\pm$0.0261  |
> | 6           | 0.0222$\pm$0.0016  | 3241.17$\pm$44.89  | 0.6676$\pm$0.0334  |
> | 8           | 0.0204 $\pm$0.0017 | 2911.97$\pm$58.3  | 0.6282$\pm$0.0358  |
> | $\infty$     | 0.0194 $\pm$0.0018  | 2798.53$\pm$29.11  | 0.6110 $\pm$0.0292  |
>
> We note that no official implementation for Badanidiyuru et al. (2023) is available. Our reproduction of their method can be found at the following link: https://github.com/anonymousaabb/Regression_Privacy.
>
> **W8.2**: Second, the experiments are conducted on only one simple real-world dataset. Compared to existing works like Badanidiyuru et al., 2023, the experimental investigation is inadequate.
>
> The authors should compare their method to Badanidiyuru et al., 2023 on more datasets.
>
> **A**: To clarify, our numerical experiments were performed on the following datasets: Communities and Crime, Criteo Sponsored Search Conversion Log, and California Housing.

---

> > ### Comment · Reviewer_nRv3 · 2025-11-24
> >
> > Thank you for the clarification. The hyperparameter issue is still confusing. For example, the learning rate and regularization techniques (e.g., early stopping, weight decay) are important to the training procedure but were not specified in the rebuttal. According to Appendix G of Badanidiyuru et al., 2023, the details of applying the "unbiased" method to the US Census and Criteo datasets can be found. Under their hyperparameters, for example, the testing loss for $\epsilon=0.3$ is only 5000-6000, which is much smaller than the results shown by the authors in their rebuttal table. Thus, the comparison seems unfair. Considering the effort the authors made to clarify my other concerns, I can revise my rating to a 4.

---

> > > ### Author Response · Authors · 2025-11-24
> > > **Response to Reviewer nRv3**
> > >
> > > Dear Reviewer nRv3,
> > >
> > > Thank you for your constructive feedback and for raising your score for our paper. We apologize for any confusion.
> > >
> > > **Q1**: The hyperparameter issue is still confusing. For example, the learning rate and regularization techniques (e.g., early stopping, weight decay) are important to the training procedure but were not specified in the rebuttal.
> > >
> > > **A**: The hyperparameters are as follows (also see Appendix H in revision):
> > >
> > > - **Crime**: We train it using the Adam optimizer with an initial learning rate of $0.001$. The model undergoes training for a total of 50 epochs and set the batchsize equal to $256$, with the learning rate decayed by a factor of 10 at the 25th epoch. For the $L_2$ regularizer, we set the factor to weight_decay $=1\text{e-}4$.
> > >
> > > - **Criteo**: We set the maximum number of epochs to $50$ and the batch size to $8192$. The model is trained using the RMSProp optimizer with an initial learning rate of $1\hbox{e-}3$, and the learning rate is decayed using the CosineAnnealingLR method. Additionally, we set the $L_2$ regularizer parameter weight_decay to $1\hbox{e-}4$.
> > >
> > > - **Housing**: The model is trained for a total of 50 epochs, with a batch size set to $256$. During training, we utilize the Adam optimizer with an initial learning rate of $0.001$. The learning rate is decayed by a factor of 10 at the 25th epoch. For $L_2$ regularization, we set the regularization factor to weight_decay $=1\text{e-}4$.
> > >
> > > In addition, there is no early stopping for all the datasets in our paper.
> > >
> > > **Q2**: According to Appendix G of Badanidiyuru et al., 2023, the details of applying the ``unbiased" method to the US Census and Criteo datasets can be found. Under their hyperparameters, for example, the testing loss for is only 5000-6000, which is much smaller than the results shown by the authors in their rebuttal table. Thus, the comparison seems unfair.
> > >
> > > **A**: Thank you for this feedback. We followed the experimental setup detailed in Appendix G of Badanidiyuru et al. (2023). We will make our best effort to address this point in this rebattal.

---

> > > > ### Author Response · Authors · 2025-11-27
> > > > **Response to Reviewer nRv3**
> > > >
> > > > Dear Reviewer nRv3,
> > > >
> > > > We are writing to thank you for your valuable feedback on our paper. We are committed to acting on the additional points you raised.
> > > >
> > > > **Numerical results** of the Criteo Sponsored Search Conversion Log Dataset for the algorithm by Badanidiyuru et al. (2023):
> > > >
> > > >
> > > > | epsilon/dataset | Criteo     |
> > > > |---------------|---------|
> > > > | 0.05         |  10572.5890$\pm$1134.12  |
> > > > | 0.1          | 7884.54$\pm$664.82  |
> > > > | 0.3          | 5807.11$\pm$393.60  |
> > > > | 0.5           | 4826.07$\pm$250.91  |
> > > > | 0.8           | 4769.94$\pm$210.42  |
> > > > | 1           | 4630.83$\pm$171.56  |
> > > > | 1.5           | 4472.40$\pm$92.37  |
> > > > | 2           | 4459.39$\pm$78.33  |
> > > > | 3             | 4330.81$\pm$58.60  |
> > > > | 4           | 4244.34$\pm$76.74  |
> > > > | 6           | 3241.17$\pm$44.89  |
> > > > | 8           | 2911.97$\pm$58.30  |
> > > > | $\infty$     | 2798.53$\pm$29.11 |
> > > >
> > > > - The numerical results are sensitive to parameters. The results will get much worse even we change the batchsize to 8192.

---

### Official Review · Reviewer_K791 · 2025-10-31

**Soundness:** 3
**Presentation:** 2
**Contribution:** 2
**Rating:** 6
**Confidence:** 2

**Summary:**

This paper studies label differentially private regression, which assumes public feature and private labels. Prior work uses randomized response on discretized labels(RR-on-Bins), and then train on noisy labels. This paper proposes to sample the noisy labels from a piecewise function, where the interval are choosing by optimizing an objective function using prior distribution. This label randomizer concentrates its mass on a neighborhood on the y, and the spread some probability mass over an optimal interval computed from prior. My understanding is the main benefit compared to prior work is to reduce the discretization-induced errors. The theoretical results is mostly on privacy proof. And the utility is demonstrated by experiments on real datasets.

**Strengths:**

1. Prior label-DP regression mechanisms typically discretize the continuous label and apply randomized response over a finite set of bins, which introduces quantization error; using a continuous non-additive randomizer directly on R removes that source of loss.
2. The paper reports consistent gains against additive baselines (Laplace/Gaussian/Staircase) and discrete non-additive baselines (RR-on-Bins and variants) at comparable epsilons, suggesting the continuous construction can be practically superior.

**Weaknesses:**

1. It’s not obvious why this is fundamentally better than RR-on-Bins: the method still depends on a prior summarized by a histogram and on selecting an interval. If the interval search effectively scans over endpoints induced by k histogram bins (potentially k^2 pairs) there’s a time–accuracy trade-off tied to k that isn’t fully analyzed. Clarifying how interval selection avoids a hidden discretization dependence in the beginning would help audience to understand this better
2. Prior work gives unbiased label randomizers and studies bias–variance trade-offs. this paper acknowledges it does not enforce unbiasedness

**Questions:**

1. I am confused by the claim that this is first non-additive mechanism. Prior label-DP regression already used non-additive RR-on-Bins. Is the novelty specifically “continuous non-additive”?

2. Beyond the qualitative argument “no quantization,” can you give a formal or empirical decomposition (e.g., bias from binning + stochastic error) showing when continuous non-additive strictly dominates additive noise-adding mechanisms and RR-on-Bins?

3. For highly skewed/long-tailed label distributions (e.g., ad-click or revenue labels), do you have distributional assumptions under which your mechanism yields a provable MSE (or regret) advantage over RR-on-Bins?

---

> ### Author Response · Authors · 2025-11-15
> **Reply to Reviewer K791**
>
> Dear Reviewer K791,
>
> We thank the reviewer for providing the theoretical suggestions for comparisons with some existing method.
>
> **W1** It’s not obvious why this is fundamentally better than RR-on-Bins: the method still depends on a prior summarized by a histogram and on selecting an interval. If the interval search effectively scans over endpoints induced by $k$ histogram bins (potentially $k^2$ pairs) there’s a time–accuracy trade-off tied to $k$ that isn’t fully analyzed. Clarifying how interval selection avoids a hidden discretization dependence in the beginning would help audience to understand this better.
>
> **A**: We provide a comparison of time. The time for the methods are almost the same except RRonBins. In the following we report the time (seconds) over 10 trails for all the methods when $\epsilon=8$. Similar results for other $\epsilon$ and datasets:
> | Datasets/Method|Laplace|Gaussian|Staircase|RRonBins| Our| Unbiased|
> |---------------|---------|---------|---------|---------|---------|---------|
> |Housing| 698.82|698.7013|700.1707|768.3160|700.7209|700.1371|
>
> **Discretization in RRonBins**:
> In our implementation, we discretize the original responses by applying torch.round(y, decimals=2). We found that using a larger *decimals* resulted in high computational cost, due to the use of dynamic programming. The total runtime for the experiment was approximately 1 hour and 43 minutes when decimals=3.
>
> **Unbiased**:
> For the unbiased mechanism proposed by Badanidiyuru et al. (2023), we set the size of both the original and randomized response spaces to 5. The computational cost of the mechanism is highly sensitive to the cardinality of the original and randomized response spaces.
>
> In addition, we analyze the differences of regression and classification problems as follows:
>
> In machine learning theory, **regression** and **classification** are distinct tasks, defined by their output spaces:
>
> **Regression** addresses estimation problems where the output space $Y$ is **continuous** and ordered, with the goal of predicting a real-valued quantity.
>
> **Classification** deals with problems where the output space is a **finite** set of **discrete** categories.
>
> The methods of Ghazi et al. (2022) and Badanidiyuru et al. (2023) fundamentally **convert a regression problem into a classification problem** within the framework of Label Differential Privacy. For RRonbins, this conversion is achieved through a multi-step process:
>
> **Rounding**: The original continuous responses (e.g., the Criteo Search dataset) are first rounded to integers.
>
> **Discretization (Binning)**: A dynamic programming algorithm is used to partition the rounded values into a finite set of bins, effectively creating a set of discrete classes.
>
> **Randomization**: A standard Randomized Response (RR) mechanism is applied over this finite set of bins.
>
> In contrast, our method **directly** addresses the regression problem under Label DP.
>
> **W2**: Prior work gives unbiased label randomizers and studies bias–variance trade-offs. this paper acknowledges it does not enforce unbiasedness.
>
> **A**: As mentioned at Q4 by Rewiewer dkUH, there is no rigorous definition of unbiased randomizers (used in Badanidiyuru et al., 2023), although we do not extend our method to the so-called unbiasedness.
>
> In classical statistics, an estimator is unbiased if its expected value equals the true parameter value, i.e., $\mathbb{E}[\hat{\theta}] = \theta$. A natural extension for a randomized mechanism $\mathcal{M}$ would be $\mathbb{E}[\mathcal{M}(y)] = y$ for all $y \in \mathcal{Y}$. However, a rigorous and universally adopted definition of unbiasedness within the framework of differential privacy is currently lacking.
>
> **Q1**: I am confused by the claim that this is first non-additive mechanism. Prior label-DP regression already used non-additive RR-on-Bins. Is the novelty specifically “continuous non-additive”?
>
> **A**: We apologize for any confusion. To clarify, our contribution is the first introduction of **continuous** random variables for constructing non-additive mechanisms in label differential privacy.

---

> ### Author Response · Authors · 2025-11-15
> **Response to Reviewer K791**
>
> **Q2**: Beyond the qualitative argument “no quantization,” can you give a formal or empirical decomposition (e.g., bias from binning + stochastic error) showing when continuous non-additive strictly dominates additive noise-adding mechanisms and RR-on-Bins?
>
> **A**: Thank you for the help suggestions. In the following we analyze the error bound of RRonBins. We divide the error bound into two: estimation error ($E_1$) and stochastic error ($E_2$). When we use the same neural networks in training, the bound of $E_1$ is the same, while $E_2$ is larger due to the introduction of discretizations from continuous respons.
>
> The details of analysis is as follows. Let $\Omega=[0,1]^d$ and $\ell(\cdot,\cdot)$ be $L_p$ norm, $x,y,\tilde y$ be a predictor, a true response and a randomized response, respectively. Let $\tilde g(x)$ be a predicted regression function by $(x,\tilde y)$ and $g(x)\in L_p(\Omega)$ be a true regression function such that $g(x)=\tilde y$. Then the error can be bounded as
> $$\mathbb E_{x,y,\tilde y}[\ell(\tilde g(x),y)]\le \mathbb E_{x,y,\tilde y}[\ell(\tilde g(x),\tilde y)]+\mathbb E_{x,y,\tilde y}[\ell(\tilde y,y)]:=E_1+E_2.$$
>
> In deep learning, $E_1$ (model risk) decays as network complexity increases. From [1], we have
>
> $$\mathbb E_{x,y,\tilde{y}}[\ell(\tilde{y},y)]=\mathbb E_{y,\tilde{y}}[\ell(\tilde{y},y)]=\mathbb E_y[\mathbb E_{\tilde y|y}\ell(\tilde y,y)].$$
>
> To get the randomized responses, the RRonBins algorithm need the following three steps:
>
> - Round the original responses (Criteo Search dataset to integers)
>
> - Find finite bins by dynamic programming
>
> - Randomization by RR
>
> The first two steps correspond to finding optimal finite bins. For an original response, let $\hat y=\hat y(y)$ be the modified response after rounding, $\breve{y}=\breve{y}(y)$ be the response corresponding to finite bins.
>
> In short, The authors discrete the continuous response $\mathcal{Y}$ to discrete $\tilde{\mathcal{Y}}=\lbrace\breve y_1,\cdots,\breve y_N\rbrace$, then randomize $\hat{\mathcal{Y}}$ by RR. The value of $E_2$ should be
>
> $$E_2=\mathbb E_y[\ell(\breve{y},y)]+\mathbb E_{\breve{y}}[\mathbb E_{\tilde{y}|\breve{y}}\ell(\tilde y,\breve{y})]\ge \mathbb E_y[\mathbb{E}_{\tilde y|y}\ell(\tilde y,y)],$$
> where the last inequality is due to the introduction of extra variables when discretization from continuous variables.
>
> [1] J. W. Siegel. Optimal approximation rates for deep relu neural networks on sobolev and besov spaces. Journal
> of Machine Learning Research, 24(357):1–52, 2023.
>
> **Q3**: For highly skewed/long-tailed label distributions (e.g., ad-click or revenue labels), do you have distributional assumptions under which your mechanism yields a provable MSE (or regret) advantage over RR-on-Bins?
>
> **A**: We consider the underlying data exhibits a highly skewed, long-tailed distribution.
>
> The core objective in differential privacy (DP) is to maximize the accuracy of queries or models while satisfying the privacy guarantee. Standard Randomized Response (RR) mechanisms are inherently most effective for balanced datasets, as they preserve the relative frequency of all classes equally.
>
> However, in the context of a **long-tailed** data distribution, this approach is suboptimal. The infrequent items in the tail contribute minimally to the overall utility, but consume a disproportionate amount of the privacy budget. We posit that for such skewed distributions, superior overall performance can be achieved by strategically **ignoring the most infrequent data points** and concentrating the privacy budget on preserving the accuracy of the more common ones.
>
> Our method embodies this philosophy by designing an optimal interval. This interval controls the subset of the data where the randomized responses are.
>
> We acknowledge that a direct theoretical comparison with bin-based RR methods (RRonBins) is highly challenging:
>
> - RRonBins operates on a **discrete** space of finite bins.
>
> - Our proposed method operates on a **continuous** interval.
>
> These represent two distinct mathematical frameworks for achieving privacy. We believe a rigorous theoretical analysis bridging these two approaches is a valuable and non-trivial research direction, which we plan to pursue in future work.

---

> > ### Comment · Reviewer_K791 · 2025-11-26
> >
> > Thanks authors for the detailed rebuttal. the clarifications and added experiments improve the paper. Some concerns remain, especially the need for a clearer theoretical justification of the method’s advantage over prior approaches. I’ve read the other reviewers’ comments and agree with several of their points. Overall, the work is interesting and empirically promising, but would benefit from stronger conceptual grounding. My score remains unchanged.

---

> > > ### Author Response · Authors · 2025-11-27
> > > **Response to Reviewer K791**
> > >
> > > Dear Reviewer K791,
> > >
> > > Thank you for your constructive feedback on our paper. We add some theoretical analysis on the algorithm by Badanidiyuru et al. (2023).
> > >
> > > **Theorem**: We consider a nonnegative response space and assume $\frac{\sum_{\hat y\in\hat{\mathcal{Y}}}\hat y-|\hat{\mathcal{Y}}|}{e^\epsilon-1}>(M-m)$, where $\hat{\mathcal{Y}}$ is the unbiased randomized space, $|\hat{\mathcal{Y}}|$ is the cardinality of $\hat{\mathcal{Y}}$ and $m,M$ are the minimum and maximum values of $\hat{\mathcal{Y}}$.  Let $\Omega=[0,1]^d$ and $\ell(\cdot,\cdot)$ be $L_p$ norm, $x,y,\tilde y$ be a predictor, a true response and a randomized response, respectively. Let $\tilde g(x)$ be a predicted regression function by $(x,\tilde y)$ and $g(x)\in L_p(\Omega)$ be a true regression function such that $g(x)=\tilde y$. Then the error can be bounded as
> > > $$\mathbb E_{x,y,\tilde y}[\ell(\tilde g(x),y)]\le 2C||f||L^{-2s/d}+\left(\frac{\sum_{\hat y\in\hat{\mathcal{Y}}}\hat{y}-|\hat{\mathcal{Y}}|}{e^\epsilon-1}\right)^2,$$
> > > where $||f||=||f||_{W^s(L_q(\Omega))}$
> > >
> > > or
> > >
> > > $||f||=||f||_{B^s_r(L_q(\Omega))}.$
> > >
> > > **Proof**:
> > > $$\mathbb E_{x,y,\tilde y}[\ell(\tilde g(x),y)]\le 2\mathbb E_{x,y,\tilde y}[\ell(\tilde g(x),\tilde y)]+2\mathbb{E}_{x,y,\tilde y}[\ell(\tilde y,y)]:=2E_1+2E_2.$$
> > >
> > > In deep learning, $E_1$ (model risk) decays as network complexity increases. From [1], we have
> > > $$E_1\le C||f||L^{-2s/d}$$
> > >  for a constant $C:=C(s,q,p,d)<+\infty$.
> > >
> > > Next, we estimate the upper and lower bounds of $E_2$. Since $\frac{\sum_{\hat y\in\hat{\mathcal{Y}}}\hat y-|\hat{\mathcal{Y}}|}{e^\epsilon-1}>(M-m)$, we have $M-\frac{\sum_{\hat y\in\hat{\mathcal{Y}}}\hat y-|\hat{\mathcal{Y}}|}{e^\epsilon-1}<m$. Note that $\mathcal{Y}$ and $\hat{\mathcal{Y}}$ share the same minimum and maximum value. That is, the minimum of the original response space $\hat{\mathcal{Y}}$ is larger than the maximum of the randomized response space $\tilde{\mathcal{Y}}$. Therefore,
> > >
> > > $$\left(\frac{\sum_{\hat y\in\hat{\mathcal{Y}}}\hat y-|\hat{\mathcal{Y}}|}{e^\epsilon-1}-(M-m)\right)^2\le(y-\tilde y)^2\le\left(\frac{\sum_{\hat y\in\hat{\mathcal{Y}}}\hat y-|\hat{\mathcal{Y}}|}{e^\epsilon-1}\right)^2$$
> > > which implies
> > > $$\left(\frac{\sum_{\hat y\in\hat{\mathcal{Y}}}\hat y-|\hat{\mathcal{Y}}|}{e^\epsilon-1}-(M-m)\right)^2\le2E_2=2\mathbb{E}(y-\tilde y)^2\le\left(\frac{\sum_{\hat y\in\hat{\mathcal{Y}}}\hat y-|\hat{\mathcal{Y}}|}{e^\epsilon-1}\right)^2$$
> > >
> > > **Note**:
> > > - The responses `ViolentCrimesPerPop` in Communities and Crime dataset, `SalesAmountInEuro` in the Criteo Sponsored Search Conversion Log Dataset and `housing price` in California Housing dataset are all nonnegative responses.
> > >
> > > - The assumption $\frac{\sum_{\hat y\in\hat{\mathcal{Y}}}\hat y-|\hat{\mathcal{Y}}|}{e^\epsilon-1}>(M-m)$ is reasonable. When $\epsilon$ is small, this assumption holds. Hence, the error bound is very large if the average of $\hat{\mathcal{Y}}$ is large.
> > >
> > > - In contrast, our algorithm choose an appropriate interval which dominates most of samples, which avoid a large error bound.
> > >
> > > [1] J. W. Siegel. Optimal approximation rates for deep relu neural networks on sobolev and besov spaces. Journal
> > > of Machine Learning Research, 24(357):1–52, 2023.

---

### Official Review · Reviewer_kXFG · 2025-10-31

**Soundness:** 2
**Presentation:** 2
**Contribution:** 2
**Rating:** 4
**Confidence:** 4

**Summary:**

This paper proposes a new mechanism RPWithPrior, for achieving $\epsilon-$label
differential privacy (LabelDP) in regression tasks. Unlike previous approaches such as
RR-on-Bins or its unbiased variants, which discretize the continuous label space before
applying randomized response (RR), the proposed method models both the original
and randomized responses as continuous random variables and determines an optimal
perturbation interval $[A_1, A_2]$ based on either known or estimated prior distributions.
The paper provides theoretical proofs that the proposed mechanism satisfies $\epsilon-$LabelDP
and validates its effectiveness on three regression datasets, showing lower mean squared
error (MSE) compared to Gaussian, Laplace, Staircase, and RR-on-Bins mechanisms.

**Strengths:**

1. The idea of formulating LabelDP regression in a continuous space rather than
discretized bins is novel.

2. Theoretical results (Propositions 3.1, Lemmas 3.3–3.5) are clearly stated, showing
that the proposed density function satisfies $\epsilon-$LabelDP.

3. The method extends naturally to the case where the prior distribution is unknown,
providing a practical histogram-based alternative.

4. Experiments across multiple datasets (Communities & Crime, Criteo, California
Housing) demonstrate consistent improvements in MSE over existing mechanisms.

**Weaknesses:**

1. The prior $f_Y (y)$ is central to the algorithm (Section 3.2) yet poorly defined. It
is unclear whether this represents: (a) an empirical label density estimated from
private data, (b) a fixed external prior, or (c) a Bayesian prior distribution. In
the case of the histogram-based extension (Section 4), privacy and accuracy both
depend critically on the quality of prior estimation. However, the paper provides
neither a sensitivity analysis with respect to estimation error nor a discussion of
how bias in the histogram affects privacy utility.

2. Theoretical derivations reproduce similar privacy constraints to Ghazi et al. (2022)
and Badanidiyuru et al. (2023). The proposed mechanism essentially replaces
discrete sampling with piecewise uniform densities, which still yield the same order
of privacy–utility tradeoff. There is no formal comparison or proof of improved
efficiency or optimality.

3. Algorithm 1 states an $O\left(k^2\right)$ computational complexity (Sec. 3.3), where $k$ denotes the number of candidate interval boundaries derived from the prior partition. However, $k$ is not clearly defined and may scale with the number of histogram bins or data points, which could make the practical cost much higher than implied. Moreover, Section 5 reports only accuracy metrics and omits any runtime or memory comparison with RR-on-Bins or Laplace mechanisms, so the claimed "efficiency" remains qualitative rather than quantitatively supported.

4. The experiments evaluate the method only under the uniform privacy-noise model implied by Eq. (3.4). The authors do not test how the algorithm performs when the true label or perturbation distributions deviate from uniformity (e.g., Gaussian or heavy-tailed noise). Moreover, the parameter $\delta$, which defines the perturbation range, is fixed but never analyzed or justified. Consequently, the robustness of the proposed mechanism to distributional or hyperparameter misspecification remains unclear.

**Questions:**

1. How sensitive is performance to the histogram bin width $\sigma$ in Algorithm 4? Would
too coarse or too fine bins degrade utility or violate privacy due to poor prior
approximation?

2. How is the total privacy budget composed between Laplace prerandomization and
RPWithPrior sampling? Is it strictly additive $(\epsilon_1 + \epsilon_2)$ or governed by a tighter
composition theorem?

3. Can the method extend to multivariate regression or non-scalar labels, and if so,
how would the conditional density generalize?

4. Have you considered comparing to Badanidiyuru et al. (2023) empirically to assess
unbiasedness and efficiency directly?

---

> ### Author Response · Authors · 2025-11-15
> **Response to Reviewer kXFG**
>
> Dear Reviewer kXFG,
>
> Thank you for your thoughtful review and constructive comments on our paper.
>
> **W1.1**: The prior $f_Y (y)$ is central to the algorithm (Section 3.2) yet poorly defined. It is unclear whether this represents: (a) an empirical label density estimated from private data, (b) a fixed external prior, or (c) a Bayesian prior distribution.
>
> **A**: Our randomization procedure is designed to handle both scenarios where the prior distribution of the data is known and unknown. The overall approach is a two-phase process when the prior is unknown: first estimate, then randomize.
>
> **Known Prior** When the prior distribution of the data, $P$, is known, we apply Algorithm 2 directly.
>
> **Unknown Prior** In the more common scenario where the prior $P$ is unknown, we propose a two-stage method outlined in Algorithm 3:
>
> **Stage 1: Prior Estimation**. We first obtain an estimate of the prior distribution, denoted as $\hat P$. This is achieved using Algorithm 4, which employs a histogram-based approach on a subset of the data.
>
> **Stage 2: Plug-in Randomization**. The estimated prior $\hat P$ is then used as input to the core Algorithm 2.
>
> **W1.2**: In the case of the histogram-based extension (Section 4), privacy and accuracy both depend critically on the quality of prior estimation. However, the paper provides neither a sensitivity analysis with respect to estimation error nor a discussion of how bias in the histogram affects privacy utility.
>
> **A**:  The performance of our method is largely insensitive to the value of $\sigma$. In empirical tests, varying $\sigma$ by a factor of $\alpha$ around the fixed values $\lbrace0.5, 1, 2\rbrace$ resulted in negligible change to the outcome. This robustness is expected, as Algorithm 4 is used solely to estimate the prior distribution for determining the optimal interval.
>
> **W2**: Theoretical derivations reproduce similar privacy constraints to Ghazi et al. (2022) and Badanidiyuru et al. (2023). The proposed mechanism essentially replaces discrete sampling with piecewise uniform densities, which still yield the same order of privacy–utility tradeoff. There is no formal comparison or proof of improved efficiency or optimality.
>
> **A**: In machine learning theory, **regression** and **classification** are distinct tasks, defined by their output spaces:
>
> **Regression** addresses estimation problems where the output space $Y$ is **continuous** and ordered, with the goal of predicting a real-valued quantity.
>
> **Classification** deals with problems where the output space is a **finite** set of **discrete** categories.
>
> The methods of Ghazi et al. (2022) and Badanidiyuru et al. (2023) fundamentally **convert a regression problem into a classification problem** within the framework of Label Differential Privacy.
> In contrast, **our method directly** addresses the regression problem under Label DP.
>
> **W3**: Algorithm 1 states an $O\left(k^2\right)$ computational complexity (Sec. 3.3), where $k$ denotes the number of candidate interval boundaries derived from the prior partition. However, $k$ is not clearly defined and may scale with the number of histogram bins or data points, which could make the practical cost much higher than implied. Moreover, Section 5 reports only accuracy metrics and omits any runtime or memory comparison with RR-on-Bins or Laplace mechanisms, so the claimed "efficiency" remains qualitative rather than quantitatively supported.
>
> **A**: The time for the methods are almost the same except RRonBins. In the following we report the time (seconds) over 10 trails for all the methods when $\epsilon=8$. Similar results for other $\epsilon$ and datasets:
> |Datasets/Method|Laplace|Gaussian|Staircase|RRonBins|Our|Unbiased|
> |-------|-------|-------|-------|-------|-------|-------|
> |Housing|698.82|698.7013|700.1707|768.3160|700.7209|700.1371|
>
> **Discretization in RRonBins**:
> In our implementation, we discretize the original responses by applying torch.round(y, decimals=2). We found that using a larger *decimals* resulted in high computational cost, due to the use of dynamic programming. The total runtime for the experiment was approximately 1 hour and 43 minutes when decimals=3.
>
> **Unbiased**:
> For the unbiased mechanism proposed by Badanidiyuru et al. (2023), we set the size of both the original and randomized response spaces to 5.
>
> Here we also report the time and space complexities of their differences
> - Compute the optimal $\Phi$ (Algo. 2 in Ghazi et al. (2022)),
> - Compute Optimal Interval (Algo. 1 of ours),
> - Compute OptUnbiasedRand$_\epsilon$ (Algo. 1 of Badanidiyuru et al. (2023)).
>
> The results are in the following table:
> |Complexity/Method|RRonbins|Our|Unbiased|
> |-------------|------|------|------|
> | Time| $O(n^3_1)$|$O(n^2_2)$|$O(n^{2.5}_3 n^{2.5}_4)$|
> | Space| $O(n^2_1)$|$O(n_2)$| $O(n_3 n_4)$|
>
> where $n_1$ is the unique number of discretized samples, $n_2$ is the number of intervals and $n_3,n_4$ are the numbers of the original and randomized spaces.

---

> ### Author Response · Authors · 2025-11-15
> **Reply to Rewiewer KxFG**
>
> **W4.1**: The experiments evaluate the method only under the uniform privacy-noise model implied by Eq. (3.4). The authors do not test how the algorithm performs when the true label or perturbation distributions deviate from uniformity (e.g., Gaussian or heavy-tailed noise).
>
> **A**: The optimization problem posed in Eq. (3.3) is central to our mechanism. As stated in Lemma 3.3, Eq. (4.3) [formerly Eq. (3.4)] provides the closed form for this problem under the specific assumption that $f_{\tilde Y|Y}(\tilde y,y), \forall y\in\mathcal I,\forall\tilde y\in \mathcal N_y$.
> This assumption is key to deriving a tractable, closed-form solution. For other, more general assumptions on the form of $f_{\tilde Y|Y}(\tilde y,y)$, the optimization problem (3.3) becomes significantly more complex.
>
> **W4.2**: Moreover, the parameter $\delta$, which defines the perturbation range, is fixed but never analyzed or justified. Consequently, the robustness of the proposed mechanism to distributional or hyperparameter misspecification remains unclear.
>
> **A**: The value of $\delta$ for different datasets (also see Appendix G.4):
>
> | dataset/$\epsilon$ | 0.05    | 0.1     | 0.3     | 0.5     | 0.8  | 1| 1.5| 2| 3| 4| 6 | 8 |$\infty$|
> |---------------|---------|---------|---------|---------|---------|---------|---------|---------|---------|---------|---------|---------|---------|
> | Crime         | 0.4  | 0.4  | 0.4  | 0.4  | 0.4  | 0.4  | 0.4  | 0.4  | 0.4  | 0.4  | 0.4  | 0.4  |0.4  |
> | Criteo         | 70  | 70  | 70  | 70  | 70  | 70  | 70  | 70  | 70  | 70  | 70  | 70  |70  |
> | Housing        | 0.7  | 0.5  | 1.2  | 1.0  | 2.2  | 1.5  | 1.5  | 1.5  | 1.4  | 1.2  | 0.7  | 0.1  |0.1  |
>
> We choose parameters manually, similar as grid search.
>
> **Q1**: How sensitive is performance to the histogram bin width $\sigma$ in Algorithm 4? Would too coarse or too fine bins degrade utility or violate privacy due to poor prior approximation?
>
> **A**: The performance of our method is largely insensitive to the value of $\sigma$. In empirical tests, varying $\sigma$ by a factor of $\alpha$ around the fixed values ${0.5, 1, 2}$ resulted in negligible change to the outcome. This robustness is expected, as Algorithm 4 is used solely to estimate the prior distribution for determining the optimal interval.
>
> Furthermore, we observe a clear empirical relationship between the privacy budget and the optimal interval: a smaller privacy budget $\epsilon$ leads to a smaller optimal interval, and vice versa.
>
> **Q2**: How is the total privacy budget composed between Laplace prerandomization and RPWithPrior sampling? Is it strictly additive $\epsilon_1+\epsilon_2$ or governed by a tighter composition theorem?
>
> **A**:
> - **Interval Determination**: In the first phase, we use a Laplace prerandomization mechanism to determine the optimal interval in a differentially private manner.
>
> - **Sampling**: In the second phase, we use the RPWithPrior algorithm to generate the final privatized responses, conditioned on the optimal interval from the first phase.
>
> The privacy guarantee for the entire process is provided by the sequential composition theorem of differential privacy. Since the two phases operate on the same dataset, the total privacy cost is the sum of the privacy budgets allocated to each phase.
>
> **Q3**: Can the method extend to multivariate regression or non-scalar labels, and if so, how would the conditional density generalize?
>
> **A**: Yes, the framework can be extended to the multivariate case. The most straightforward approach is to assume the perturbation mechanisms across dimensions are independent. Formally, we assume the conditional density factorizes as:
> $$f(\tilde{y}|y)=\Pi^k_{i=1}f_i(\tilde{y}_i|y_i).$$
> Under this assumption, we can apply a one-dimensional perturbation mechanism to each dimension $y_i$ separately and independently. This principle of independent noise across dimensions is, in fact, the standard approach used by additive noise mechanisms like the multivariate Laplace or Gaussian mechanism.

---

> ### Author Response · Authors · 2025-11-22
> **Reply to Rewiewer KxFG**
>
> **Q4**: Have you considered comparing to Badanidiyuru et al. (2023) empirically to assess unbiasedness and efficiency directly?
>
> **A**: We provide the comparison results as follows:
>
> | epsilon/dataset | Crime    | Criteo     | Housing     |
> |---------------|---------|---------|---------|
> | 0.05         | 0.1293$\pm$0.0096  | 10572.59$\pm$1134.12  | 1.6307$\pm$0.1155  |
> | 0.1          | 0.1194$\pm$0.0056  | 7884.54$\pm$664.82  | 1.6554$\pm$0.0542  |
> | 0.3          | 0.1110$\pm$0.0094  | 5807.11$\pm$393.60  | 1.5656$\pm$0.0625  |
> | 0.5           | 0.1037$\pm$0.0089  | 4826.07$\pm$250.91  | 1.4666$\pm$0.0644  |
> | 0.8           | 0.0892$\pm$0.0064  | 4769.9424$\pm$210.42  | 1.42419$\pm$0.0700  |
> | 1           | 0.0826$\pm$0.0039  | 4630.8307$\pm$171.56  | 1.3641$\pm$0.0617  |
> | 1.5           | 0.0668$\pm$0.0041  | 4472.4029$\pm$92.37  | 1.1896$\pm$0.0447  |
> | 2           | 0.0521$\pm$0.0032  | 4459.39$\pm$78.33  | 1.0882$\pm$0.0657  |
> | 3             | 0.0355$\pm$0.0021  | 4330.81$\pm$58.60  | 0.9587$\pm$0.0766  |
> | 4           | 0.0259$\pm$0.0024  | 4244.34$\pm$76.74  | 0.7659$\pm$0.0261  |
> | 6           | 0.0222$\pm$0.0016  | 3241.17$\pm$44.89  | 0.6676$\pm$0.0334  |
> | 8           | 0.0204 $\pm$0.0017 | 2911.97$\pm$58.30  | 0.6282$\pm$0.0358  |
> | $\infty$     | 0.0194 $\pm$0.0018  | 2798.53$\pm$29.11  | 0.6110 $\pm$0.0292  |
>
>
> We note that no official implementation for Badanidiyuru et al. (2023) is available. Our reproduction of their method can be found at the following link: https://github.com/anonymousaabb/Regression_Privacy.

---

### Official Review · Reviewer_4Cud · 2025-10-31

**Soundness:** 2
**Presentation:** 2
**Contribution:** 2
**Rating:** 4
**Confidence:** 3

**Summary:**

The paper studies the problem of optimizing randomized response mechanisms over a continuous domain $\mathcal{Y}$ under both known and unknown prior settings, with applications to regression under label-level differential privacy.

In the known prior setting, the paper assumes a prior distribution over $\mathcal{Y}$ whose density function $f_Y(y)$ is piecewise constant.
Based on this assumption, it formulates an optimization problem for designing a “good” differentially private perturbation mechanism and proposes an algorithm to find the optimal solution.
The algorithm runs in time quadratic in the number of pieces of the piecewise constant utility function.

In the unknown prior setting, the paper proposes to first estimate the prior (as a piecewise constant density function). However, the privacy and utility guarantees in this section are not clearly stated or analyzed.

**Strengths:**

The proposed optimization problem admits an optimal solution, and the paper presents a polynomial-time algorithm to compute it.

**Weaknesses:**

The paper is not well written.

Moreover, the privacy and utility guarantees of some of the proposed algorithms are unclear and cannot be  verified from the current presentation.

**Questions:**

In Section 4, the paper proposes using $\mathcal{M}_{\epsilon, \text{Lap}}(y)$ to protect $y \in \mathcal{Y}$.
However, this part is not clearly explained:

1. What is the size or structure of $\mathcal{Y}$?
2. How large can $|y - y'|$ be, where $y'$ is obtained by replacing $y$ with its counterpart in a neighboring dataset?
   Specifically, can $y'$ be any arbitrary data point in $\mathcal{Y}$?
   If $\mathcal{Y} = \mathbb{R}$ and $y'$ can take any value in $\mathbb{R}$, then the Laplace mechanism would require infinite variance, resulting in meaningless utility guarantees.

**Additional comments:**

3. In line 253, the reference to Equation (E.1) appears to be a typo—it should likely refer to Equation (3.6).

4. It is unclear how the parameter $\delta$ is chosen in Lemma 3.3.

5. It would strengthen the paper to include a clearer motivation for optimizing the objective function in Section 3.1.
   The paper provides details on *how* to perform the optimization, but offers little explanation of *why* this formulation is meaningful.
   For example, what is the underlying rationale for optimizing $F(A_1, A_2)$, and how does it connect to the overall utility guarantee of the mechanism?

---

> ### Author Response · Authors · 2025-11-15
> **Response to Reviewer 4Cud**
>
> Dear Reviewer 4Cud,
>
> Thank you for reviewing our paper and for the feedback.
>
> **Q1**: In Section 4, the paper proposes using $\mathcal{M}_{\epsilon,Lap}$ to protect $y\in\mathcal{Y}$. However, this part is not clearly explained: What is the size or structure of $\mathcal{Y}$?
>
> **A**: In a regression problem, the response space $\mathcal{Y}$ is the set of all possible values for the output variable. In our paper, we assume $\mathcal{Y}$ is a bounded set, for example, $[A_1,A_2] \subset \mathbb{R}$.
>
> **Q2**: How large can $|y-y'|$ be, where $y'$ is obtained by replacing $y$ with its counterpart in a neighboring dataset? Specifically, can $y'$ be any arbitrary data point in $\mathcal{Y}$? If $\mathcal{Y}=\mathbb{R}$ and $y'$ can take any value in $\mathbb{R}$, then the Laplace mechanism would require infinite variance, resulting in meaningless utility guarantees.
>
> **A**: In additive noise mechanisms (for example Laplace mechanism), $|y-y'|$ is dependent to the sensitivity, which is defined as follows:
>
> **Definition** [Global sensitivity]
> Let $f$ be a query function. The global sensitivity $\Delta(f)$ of $f$ between two neighboring datasets $D$ and $D'$ is defined as follows:
> $$\Delta(f)=\max||f(D)-f(D')||_1.$$
>
> In Laplace mechanism, $\lambda=\frac{\Delta}{\epsilon}$, where $\Delta$ is a sensitivity and $\epsilon$ is a privacy budget.
>
> **Q3**: In line 253, the reference to Equation (E.1) appears to be a typo—it should likely refer to Equation (3.6).
>
> **A**: Thank you for the helpful comment.
>
> **Q4**: It is unclear how the parameter $\delta$ is chosen in Lemma 3.3.
>
> **A**: The value of $\delta$ for different datasets are as follows (also see Appendix G.4):
>
> | dataset/$\epsilon$ | 0.05    | 0.1     | 0.3     | 0.5     | 0.8     | 1       | 1.5     | 2       | 3       | 4       | 6       | 8       |$\infty$       |
> |---------------|---------|---------|---------|---------|---------|---------|---------|---------|---------|---------|---------|---------|---------|
> | Crime         | 0.4  | 0.4  | 0.4  | 0.4  | 0.4  | 0.4  | 0.4  | 0.4  | 0.4  | 0.4  | 0.4  | 0.4  |0.4  |
> | Criteo         | 70  | 70  | 70  | 70  | 70  | 70  | 70  | 70  | 70  | 70  | 70  | 70  |70  |
> | Housing        | 0.7  | 0.5  | 1.2  | 1.0  | 2.2  | 1.5  | 1.5  | 1.5  | 1.4  | 1.2  | 0.7  | 0.1  |0.1  |
>
> We choose parameters manually, similar as grid search.
>
> From the above table, we find it is not sensitive of the parameter $\delta$.
>
> **Q5.1**: It would strengthen the paper to include a clearer motivation for optimizing the objective function in Section 3.1.
>
> **A**: Our method is motivated by the empirical distribution of the responses. Since most responses are concentrated within a finite interval, with only a few extreme outliers, we seek an optimal interval $\mathcal{I}$. This interval maximizes the probability that a response $y$ is mapped to a noisy value $\tilde{y}$ within its neighborhood $\mathcal{N}_y$, for all $y \in \mathcal{I}$, while still satisfying $\epsilon$-label differential privacy.
>
> **Q5.2**: The paper provides details on how to perform the optimization, but offers little explanation of why this formulation is meaningful. For example, what is the underlying rationale for optimizing $F(A_1,A_2)$, and how does it connect to the overall utility guarantee of the mechanism?
>
> **A**: The objective of optimization problem (3.3) is to ensure the highest probability that the response $y$ can be mapped to its neighborhood $[y-\delta,y+\delta]$ without violating $\epsilon$-label differential privacy.
>
> For a given interval $[A_1, A_2]$, we use $F(A_1, A_2)$ denote the maximum achievable value of the objective function in (3.3) (the derivation is in Lemma 3.3 for revision). Our goal is to find the optimal interval by maximizing $F(A_1, A_2)$ over all possible $A_1$ and $A_2$.
>
> Therefore, the mechanism maximizing the probability which maps original responses to their neighborhoods ensures high utility, while ensuring $\epsilon$-label differential privacy.

---

### Author Response · Authors · 2025-12-02
**Reply to ACs**

Dear ACs,

Thank you for overseeing the review process for our paper. We are also grateful to the reviewers for their thoughtful and constructive feedback.

## **Summary of reviewer's comments**:

[4Cud] The reviewer raises a question regarding

- the **cardinality** of the original response space, especially for **Laplace** mechanism,
- **motivations**,
-  **parameter and the optimal interval selection**.

[kXFG] This reviewer's comments mainly focus on:

- **Parameters**: Concerns about the Prior Estimation and neighborhood parameters and their stabilities.

- **formal comparison**: comparison with Ghazi et al. (2022) and Badanidiyuru et al. (2023).

- **Complexity \& Numerical Experiments**: time/space complexity and comparison with the method by Badanidiyuru et al. (2023).

[K791] This reviewer provides some **theoretical comments for comparisons** with some existing methods.

[nRv3] This reviewer mainly focus on some details of this paper, including the

- **Motivations**,

- **Details of optimization model and solution**,

- **Misunderstanding about continuous random variables**,

- **Numerical comparisons**.

[dkUH]
- **Theoretical Analysis and Empirical Comparisons** with existing methods.

- **Complexity Analysis**: including time/space complexity.

- **Sensitivity Study**: Investigation into the robustness of the method with respect to its key parameters.

## Based on the comments of all reviewers, we have updated as follows (Marked in red at revision):

1. Clarify the cores of the existing methods by Ghazi et al. (2022) and Badanidiyuru et al. (2023) (Lines 59-60) and add the motivations of our paper (Lines 188-191).

2. Add numerical comparisons with the method by Badanidiyuru et al. (2023) in Section 5.

3. Add theoretical and numerical comparisons in the appendix (see the roadmap of appendices blow).

4. Reformulate the limitations of this work: including an extension to multivariate regression or non-scalar labels and investigation of unbiasedness.

## **Roadmap of appendix**:

1. move those definitions to Appendix A as a preliminary.

2. Report the parameters used in experiments and their robustness in Appendix H.4.

3. Add numerical experiments about simulated datasets (linear or nonlinear) in Appendix I.

4. Add comparisons between some existing methods for regression and ours, including the differences of the standpoints, theoretical comparison and time/space complexity in Appendix J.

We thank PCs, SACs, ACs and all the reviewers for the opportunity to rebuttal our paper.

Best,

Authors

---

### Meta-Review · Area_Chair_JQqH · 2026-01-12

**Summary:**

This paper proposes RPWithPrior, a method for regression under label differential privacy that incorporates prior information into the randomized response mechanism. The goal is to improve utility by leveraging prior knowledge while maintaining formal privacy guarantees. The paper presents a principled formulation, provides theoretical analysis, and includes empirical results demonstrating improved performance over baseline approaches that do not use priors.

While the idea of incorporating prior information into label-DP mechanisms is reasonable and potentially useful, I do not believe the paper meets the acceptance bar for ICLR in its current form. Several reviewers raised concerns about clarity, positioning, and novelty relative to existing label-DP and discretization-based approaches. In particular, it remains unclear to what extent the proposed method offers a fundamentally new perspective versus a refinement of known techniques, and whether the empirical gains are robust across realistic prior misspecification scenarios. Although the rebuttal is careful and responsive, it does not fully resolve concerns about the strength of the contribution or its relevance to a broad machine learning audience. I therefore recommend rejection.

**Reviewer Concerns:**

Some concerns were addressed by the rebuttal. K791 asked for clarification on the assumptions behind the prior and the intended scope of applicability, and the rebuttal helped clarify that the method is primarily designed for settings where reasonable prior information is available. 4Cud raised questions about technical details and presentation, several of which were clarified in the response.

However, important concerns remain outstanding. nRv3 expressed skepticism about the novelty of the approach and questioned whether incorporating priors into randomized response yields a sufficiently new contribution beyond existing label-DP techniques; this concern was not fully alleviated. dkUH raised concerns about the empirical evaluation, particularly the sensitivity of results to prior misspecification and the limited set of baselines, which remain only partially addressed. kXFG questioned the positioning of the work relative to closely related methods and whether the improvements justify the added complexity. Overall, the central concerns about novelty, robustness, and scope persist after the rebuttal.

**Reviewer Scores:**

K791 (rating 6).
This reviewer was mildly positive but explicitly indicated comfort with rejection. After discussion, the rating would likely remain unchanged or possibly decrease slightly.

4Cud (rating 4).
This reviewer found the paper reasonable but below the acceptance threshold. Clarifications in the rebuttal are unlikely to change the overall assessment.

kXFG (rating 4).
This reviewer’s concerns about positioning and contribution were only partially addressed. The rating would likely remain unchanged.

dkUH (rating 2).
This reviewer was unconvinced by the empirical and conceptual contributions. The rebuttal is unlikely to change this assessment.

nRv3 (rating 2).
This reviewer raised fundamental concerns about novelty and relevance. The rebuttal does not address these sufficiently to warrant a score increase.

Overall, while the rebuttal improved clarity, it is unlikely to have led to meaningful upward score changes across reviewers.

---

### Decision · Program_Chairs · 2026-01-26

Reject